# Implicit Neural Surface Deformation with Explicit Velocity Fields

**Lu Sang**[1,2]**, Zehranaz Canfes**[1]**, Dongliang Cao**[3]**, Florian Bernard**[3]**, Daniel Cremers**[1,2]
[1]Technical University of Munich, [2]Munich Center of Machine Learning
{lu.sang, zehranaz.canfes, cremers}@tum.de
[3]University of Bonn
{dcao, fb}@uni-bonn.de

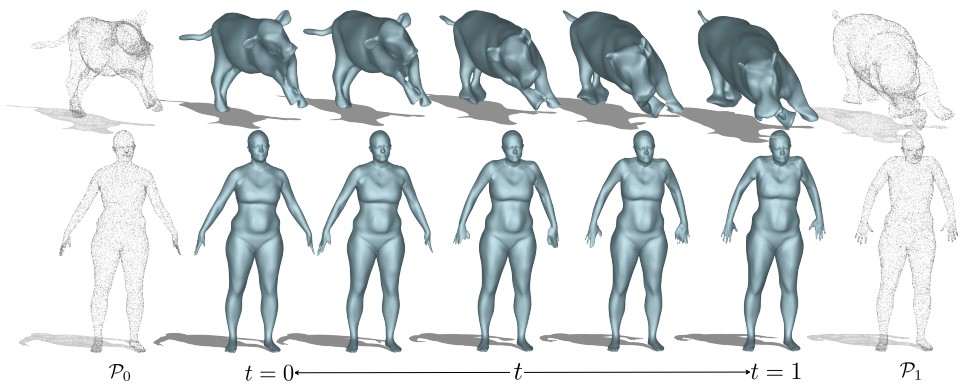

$$\mathcal{P}_0 \qquad t=0 \longleftarrow \qquad t \qquad \longrightarrow t=1 \qquad \mathcal{P}_1$$

Figure 1: Given two point clouds $\mathcal{P}_0$ and $\mathcal{P}_1$, our method predicts a time-varying neural implicit surface that represents a smooth and physically plausible deformation from $\mathcal{P}_0$ to $\mathcal{P}_1$. To ensure physical plausibility, we utilize a velocity network that leverages smoothness and divergence-free constraints.

## Abstract

In this work, we introduce the first unsupervised method that simultaneously predicts time-varying neural implicit surfaces and deformations between pairs of point clouds. We propose to model the point movement using an explicit velocity field and directly deform a time-varying implicit field using the modified level-set equation. This equation utilizes an iso-surface evolution with Eikonal constraints in a compact formulation, ensuring the integrity of the signed distance field. By applying a smooth, volume-preserving constraint to the velocity field, our method successfully recovers **physically plausible** intermediate shapes. Our method is able to handle both rigid and non-rigid deformations **without any intermediate shape supervision**. Our experimental results demonstrate that our method significantly outperforms existing works, delivering superior results in both quality and efficiency[1].

## 1 Introduction

Representing surfaces using implicit methods, such as signed distance fields, offers significant advantages over explicit methods in some applications. For example, it allows flexible topological changes and is more memory-efficient compared to storing an explicit representation of a high-resolution surface. Additionally, implicit representations allow for differentiable operations, as the respective surfaces are encoded in smooth fields, which in turn enhances a variety of downstream tasks, such as radiance field rendering by Yariv et al. (2021); Wang et al. (2023). Embedding a signed distance field within a neural network to represent a single surface demonstrated many successful outcomes, such as work from Sitzmann et al. (2020); Gropp et al. (2020); Mescheder et al. (2018). However, using implicit representations to model surface deformation or a dynamic surface evolution, especially with

---

[1]the code is available: https://github.com/Sangluisme/Implicit-surf-Deformation

physically plausible deformations, still remains challenging. The challenges stem mainly from two inherent characteristics with implicit methods: (i) implicit representations do not store explicit surface point locations, which makes it hard to directly manipulate surfaces during deformation. (ii) the lack of traceable neighboring information in implicit fields prevents the use of efficient physical constraints, for example, as-rigid-as-possible regularisation, proposed by Sorkine & Alexa (2007), which is crucial in many mesh-based methods such as the work from Alexa et al. (2023); Eisenberger et al. (2021); Cao et al. (2024a). In this paper, we aim to tackle these core problems of implicit surface representations. To this end, we introduce a method that simultaneously recovers implicit neural representations of two given point cloud inputs, together with time-varying intermediate shapes between them. Most notably, our approach distinguishes itself from previous deformation methods based on implicit representations by recovering physically plausible intermediate shapes – without supervision from ground truth intermediate shapes. To achieve this goal, we model the deformation of surface points by training a velocity network that utilizes smoothness and divergence-free constraints, thereby ensuring natural and physically plausible deformations. Our approach circumvents the need for mesh rendering during training, facilitating an end-to-end and fully differentiable training process. Our method supports both intrinsic and extrinsic deformations of the given point clouds, enhancing its versatility and application scope. In summary, we claim the following contributions:

- We propose a novel end-to-end framework that recovers the underlying surfaces of given point clouds together with physically plausible intermediate shapes.

- Our method directly deforms the implicit field by the explicit velocity field based on the level-set equation to avoid explicit mesh rendering.

- We propose to use a modified level-set equation that combines Eikonal constraint and thereby enables a compact joint optimization while preventing degenerated signed distance fields.

- We validate our method on different datasets and demonstrate that our methods give rise to high-quality interpolations for challenging inputs, both quantitatively and qualitatively.

## 2 RELATED WORKS

**Surface representation methods**   We roughly divide shape representation into explicit and implicit approaches. While explicit representations, such as polygon meshes, store mesh properties, e.g. vertices, edges, and faces explicitly, implicit methods encode the surface information into function fields, such as signed distance fields (SDF). With explicit methods, it is relatively straightforward to edit the shapes, since shape properties can directly be manipulated. However, there are some drawbacks to explicit surface representations. For instance, meshes can only have a fixed topology. It is not trivial to adapt vertices and the configuration of their connections (such as edges). Implicit methods, on the contrary, allow arbitrary topological changes since no explicit surface and structural information are stored. Additionally, neural implicit representations enable arbitrary resolutions during inference, without memory increase during storage.

**Mesh-based deformation**   Mesh-based shape deformation is a well-studied problem in computer graphics. The most common strategy is to directly deform vertices based on some local deformation measurements (e.g. as-rigid-as-possible (ARAP) proposed by Sorkine & Alexa (2007), PriMo Botsch et al. (2006), etc.). Another direction is to deform intrinsic quantities like dihedral angles such as work from Alexa et al. (2023); Baek et al. (2015) before reconstructing 3D shapes. Meanwhile, other works formulate shape deformation as a time-dependent velocity field Charpiat et al. (2007); Eckstein et al. (2007) and incorporate specific constraints (e.g. volume preservation used by Eisenberger et al. (2018); Eisenberger & Cremers (2020)). Despite the great success achieved by mesh-based shape deformation methods, they rely on the local neighborhood information obtained from edges (or triangles) during deformation. Therefore, shapes have a fixed topology during deformation, which limits the applications for shapes with inconsistent topology or partiality. In contrast, our method directly works on an implicit surface, and thus has no constraint on the shape topology and is applicable for partial shapes based on spatial smoothness regularisation. In the experiment part, we demonstrate that our method is capable of deforming shapes with significant variations of shape resolution and partiality.

**Implicit-field based deformation**   Deforming implicit fields presents a significant challenge, as all information is encoded within a function field, preventing direct operations on the object. Previous

works that studied this topic are typically for physical simulation, such as the work of Osher & Paragios (2003); Museth et al. (2002); Jones et al. (2006), and these works use classical discretely stored implicit fields. More recently, the widespread adoption of neural networks to represent implicit fields in shape modeling, work fromYang et al. (2021); Sitzmann et al. (2020); Ma et al. (2020) inspired works to study shape deformation on implicit fields. Several works, e.g., Peng et al. (2021) utilizing neural networks for shape deformation have concentrated exclusively on human body movements , Chen et al. (2021); Deng et al. (2020); Božič et al. (2021) requiring prior information such as skinning details or intermediate point clouds . Others, like Cao et al. (2024b) have explored deforming implicit fields through generative or diffusion models, but these still necessitate intermediate point clouds for supervision. Some works have trained on datasets of shapes from specific object categories, Deng et al. (2021); Genova et al. (2019); Iglesias et al. (2017); Hao et al. (2020) aiming to deform from one category to another , rather than recovering plausible intermediate shapes. Some works address this problem by defining a latent space and getting a deformed shape via latent space interpolations Liu et al. (2022). Yang et al. (2021) introduced pioneering work that enables direct editing of implicit fields using user-defined handle points, ensuring the deformation remains consistent with the original object. Following this, Mehta et al. (2022) proposed using the level-set equation to deform the implicit network, providing theoretical insights into implicit field deformation. Building on this foundation, Novello et al. (2023) extended these concepts, applying them to 3D shapes, primarily focusing on smoothing surface deformations. In terms of directly deforming the implicit field using a velocity field, their work models the velocity field through linear interpolation between two pre-trained implicit networks representing the target shapes. Therefore, these works are limited to work on a pre-defined velocity field and fail to predict reasonable intermediate shapes when the deformation is not a linear translation.

In this paper, we adopt neural implicit surface representations and tackle the challenging problem of directly deforming the implicit field while recovering **physically plausible** intermediate shapes – **without any rendering or intermediate ground truth supervision**. We achieve this by modeling deformations using a velocity field and directly deforming the implicit neural network using modified level-set-equations. Moreover, our training is end-to-end without any pre-trained SDF network needed.

## 3 METHOD

Given two 3D point clouds $\mathcal{P}_0 = \{\mathbf{x}_i^0\}_i$ and $\mathcal{P}_1 = \{\mathbf{x}_i^1\}_i$, we aim to reconstruct a time-varying implicit representation of the inputs together with natural and physically plausible intermediate surfaces. To this end, we adopt a Lagrangian perspective from fluid mechanics to track the trajectory of individual points through a velocity field $\mathcal{V} : \Omega \to \mathbb{R}^3$, for $\Omega \subset \mathbb{R}^3$ being the point domain, and directly deform the time-varying implicit field $f : \Omega \times \mathcal{I} \to \mathbb{R}$, where $\mathcal{I} = [0, 1]$ is the time interval. The primary challenges are (i) modeling deformations that are realistic according to physical laws, i.e.

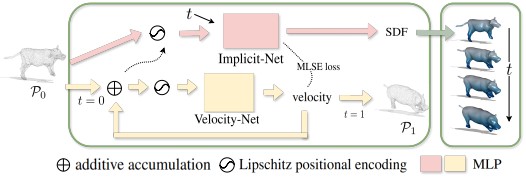

Figure 2: Pipeline of our method: given two point cloud $\mathcal{P}_0$ and $\mathcal{P}_1$, we train a time-varying Implicit-Net to predict SDF in different time steps and Velocity-Net to predict the velocity of the point at each time step. We directly deform the implicit field using MLSE loss.

modeling physically plausible movements, and (ii) deforming the implicit surfaces without relying on explicit mesh rendering or intermediate shape supervision. To tackle these issues, we employ a smoothness constraint and a divergence-free constraint on the velocity field to ensure realistic movement. Additionally, we utilize a modified level-set equation to directly deform the time-varying implicit surfaces without rendering meshes explicitly.

### 3.1 TIME-VARYING IMPLICIT FIELDS

The time-varying implicit field $f(\cdot, \cdot)$ takes a point location $\mathbf{x} \in \Omega$ and a time $t \in \mathcal{I}$ as input, where $\mathcal{I} = [0, 1]$ is the time interval and $\Omega$ is the surface domain. Its purpose is to encode the evolution of a surface, that is, the shape $\mathcal{S}_t$ at time $t$ is represented by the zero-level-set of the implicit function $f(\cdot, t)$, i.e.

$$\mathcal{S}_t = \{\mathbf{x} \in \Omega |\ f(\mathbf{x}, t) = 0\} . \tag{1}$$

Signed-distance fields (SDFs) have many outstanding properties for representing surfaces: for $C^2$-smooth surfaces, the gradient of an SDF coincides with the surface normal direction $\mathbf{n}$ on the zero-crossing contour ($\partial\Omega$), and the curvature $\kappa$ coincides with the divergence, that is

$$\mathbf{n}(\mathbf{x}, t) = \frac{\nabla f(\mathbf{x}, t)}{\|\nabla f(\mathbf{x}, t)\|} \ , \ \ \kappa(\mathbf{x}, t) = \nabla \cdot \mathbf{n}(\mathbf{x}, t), \ \text{ for } \mathbf{x} \in \partial\Omega \ . \tag{2}$$

The time-varying signed distance function $f$ should also satisfy the Eikonal equation, since at every time $t$, $f$ still is a signed distance field, as proposed by Bothe et al. (2024), i.e. at any time $t \in \mathcal{I}$,

$$\|\nabla f(\mathbf{x}, t)\| = 1 \ . \tag{3}$$

## 3.2 Velocity Fields

Inspired by the Lagrangian representation in fluid mechanics , which tracks surface points by modeling the particle trajectory $\phi : \Omega \times \mathcal{I} \to \Omega$, we track the point by estimating its velocity and integrating the velocity field to form the point trajectory. The trajectory specifically consists of the position of particle $\mathbf{x}$ at time $t$. Assuming the points are moved by an external velocity field $\mathcal{V} : \Omega \to \mathbb{R}^3$, where $\mathcal{V} \in V$, and $V$ is a Hilbert space of a smooth and compactly supported vector field on $\Omega$. The velocity of the moving particle satisfies the following ordinary differential (ODE) equation with the initial condition: the trajectory derivative w.r.t. time $t$ is the velocity and the initial point location is given by $\mathbf{x}^0$

$$\begin{cases} \mathcal{V}(\mathbf{x}) = \frac{\mathrm{d}\phi(\mathbf{x},t)}{\mathrm{d}t}, \ \text{ for } \ t \in \mathcal{I} \ , \\ \phi(\mathbf{x}, 0) = \mathbf{x}^0 \ . \end{cases} \tag{4}$$

Note that in our setting, we also enforce the ending point of particle trajectory by $\phi(\mathbf{x}, 1) = \mathbf{x}^1$, where $\mathbf{x}^1$ is the point in the target point cloud. To control the smoothness of the movement and ensure physical plausibility, we propose to constrain the velocity field by two aspects: spatial smoothness of the velocity fields, and physical constraints.

**Velocity fields that generate diffeomorphisms** As the particle path $\phi : \Omega \times t \to \Omega$ represents a trajectory from one point cloud to another, we would like to recover a smooth transformation between two point clouds, which is consistent in both directions. Thus, we seek velocity fields that generate diffeomorphisms when integrated using Eq. (4), i.e. $\phi^{-1}(\cdot, t) = \phi(\cdot, 1 - t)$

$$\phi(\mathbf{x}, t) = \mathbf{x}^0 + \int_0^t \mathcal{V}(\phi(\mathbf{x}, t))\mathrm{d}\tau \ . \tag{5}$$

Inspired by Dupuis et al. (1998a), this can be achieved by regularizing on the space $V$ through the differentiable operator $\mathcal{L} = -\alpha\Delta + \gamma\mathbf{I}$ such that

$$\|\mathcal{V}\|_V = \|\mathcal{L}\mathcal{V}\|_{l^2} = \int_\Omega \|\mathcal{L}\mathcal{V}(\mathbf{x})\|_{l^2} \, \mathrm{d}\mathbf{x} \ , \tag{6}$$

where $\mathbf{I}$ is the identity matrix. A more detailed explanation is provided in the Appendix A.

**Divergence-free velocity fields** To model physically plausible deformations, we consider the basic conservation laws. One direct conservation law we can borrow is volume conservation. Since we move points on the surface along a trajectory, we assume that no particles are moved across the surface boundary at any time. Then, the total mass inside the surface stays the same, which directly follows from the divergence theorem Kreyszig et al. (2011), i.e.

$$\nabla \cdot \mathcal{V}(\mathbf{x}) = 0. \tag{7}$$

A similar idea has also been explored in previous work from Eisenberger et al. (2018); Cosmo et al. (2020) for the case of explicit polygon meshes.

**Velocity-Net integration** Our smooth velocity field is approximated by an MLP Velocity-Net $\mathcal{V}$. To track the point trajectory, we follow the forward Euler method for integrating ODEs, i.e. we take certain discrete time steps and integrate the velocity step by step using step length $\delta T = 1/T$. Then, the discrete trajectory of points is formed by

$$\phi(\mathbf{x}, t + \delta t) = \phi(\mathbf{x}, t) + \mathcal{V}(\phi(\mathbf{x}, t))\delta t \ . \tag{8}$$

The relation between $\mathcal{P}_0$ and $\mathcal{P}_1$ is then given as $\phi(\mathbf{x}, 0) = \mathbf{x}^0$, $\mathbf{x} \in \mathcal{P}_0$ and $\phi(\mathbf{x}, 1) = \mathbf{x}^1$, $\mathbf{x} \in \mathcal{P}_1$.

### 3.3 DIRECT IMPLICIT FIELD DEFORMATION

In the previous sections, we introduced the time-varying implicit fields and velocity fields that represent the shapes and the deformation of points, respectively. In this section, we discuss how to directly deform the implicit field using the external velocity. We borrow the idea from fluid dynamics and treat every point as a fluid particle. Since points stay on the surface of any intermediate shape (i.e. $f(\phi(\mathbf{x}, t), t) = 0$ for any $t \in \mathcal{I}$ and $\mathbf{x} \in \partial\Omega$), it implies there is no in- or outflow at the surface boundary $\partial\Omega$

$$\frac{\mathrm{d}}{\mathrm{d}t} \int_\Omega f(\phi(\mathbf{x}, t), t)\mathrm{d}\mathbf{x} = 0 . \tag{9}$$

Together with the initial condition, that is, the surface deforms from the underlying surface of point cloud $\mathcal{P}_0$, Eq. (9) implies that

$$\begin{cases} \partial_t f + \mathcal{V} \cdot \nabla f = 0 \ \text{ in } \Omega \times \mathcal{I} , \\ f(\mathbf{x}, 0) = f^0 , \end{cases} \tag{10}$$

where $f^0(\mathbf{x}) = 0$ for $\mathbf{x} \in \mathcal{P}_0$. The linear transport in Eq. (10) is called the Level-Set Equation (LSE). However, the function $f$ is a signed distance function in our scenario, which means the Eikonal equation in Eq. (3) should also hold to prevent degenerated level-set functions. Previous work from Sussman et al. (1994); Sethian (1996); Sussman & Fatemi (1999) proposed to solve it by introducing a reinitialization equation at a pseudo time $\tau$, e.g. solving

$$\frac{\partial}{\partial\tau} f + \mathrm{sgn}(f^0)(\|\nabla f\| - 1), \ \ f|_{\tau=0} = f^0 . \tag{11}$$

However, this requires solving an additional partial differential equation (PDE), and requires that the signed distance field $f$ at time 0 is well initialized. To avoid pre-training a neural network to fit the starting mesh and solve the problem more compactly, we follow the idea of Bothe et al. (2024) and Fricke et al. (2022), combining Eq. (3) and

$$\frac{\mathrm{d}}{\mathrm{d}t} \|\nabla f(\mathbf{x}, t)\| = - \|\nabla f\| \langle (\nabla\mathcal{V}) \frac{\nabla f}{\|\nabla f\|} , \frac{\nabla f}{\|\nabla f\|} \rangle \equiv 0 , \tag{12}$$

with Eq. (10) to get our **Modified Level-Set Equation** (MLSE) that reads

$$\begin{cases} \partial_t f + \mathcal{V} \cdot \nabla f = -\lambda_l f \mathcal{R}(\mathbf{x}, t) \ \text{ in } \Omega \times \mathcal{I} , \\ f(\mathbf{x}, 0) = f^0 , \end{cases} \tag{13}$$

where $\mathcal{R}(\mathbf{x}, t) = -\langle (\nabla\mathcal{V}) \frac{\nabla f}{\|\nabla f\|} , \frac{\nabla f}{\|\nabla f\|} \rangle$. Our MLSE in Eq. (13) preserves the norm of the gradient at the zero-crossing contour. We also adapt the original proposed level-set equation in Bothe et al. (2024); Fricke et al. (2022) by adding a weight $\lambda_l$. We find that it helps to achieve better implicit surfaces while still preserving the desired properties. Compared to the original level-set equation Eq. (10) and discretely enforced Eikonal constraint Eq. (3) on each time step, the modified level-set equation is more compact and solves a single partial differentiable equation (PDE) in an integrated way.

Our MLSE is the bridge between the velocity field and the implicit field. Eq. (13) allows us to deform the implicit field without rendering explicit meshes. Moreover, every component of the formulation is differentiable, thus it enables end-to-end training. Our method jointly recovers the implicit surface for both given point clouds and intermediate shapes without the need for pre-training SDF neural implicit networks for given point clouds or meshes.

### 3.4 LOSS

We set up the training loss as follows. Velocity-Net loss $L_v$ contains smoothness (Eq. (6)) and divergence-free (Eq. (7)) terms. Implicit-Net loss $L_f$ contains MLSE (Eq. (13) term.

$$L_v = \int_\Omega \|\mathbf{\L}\mathcal{V}\| \, \mathrm{d}\mathbf{x} + \lambda_{\mathrm{div}} \int_\Omega |\nabla \cdot \mathcal{V}|\mathrm{d}\mathbf{x} ,$$
$$L_f = \int_\Omega |\partial_t f + \mathcal{V} \cdot \nabla f + \lambda_l f \mathcal{R}|\mathrm{d}\mathbf{x} . \tag{14}$$

The divergence-free weight $\lambda_{\mathrm{div}}$ can be set to $0$ to disable volume preservation. We show examples and the influence of the divergence-free term in the experiment section (Sec. 4).

Finally, to fit the network to the given point clouds, we propose the matching loss

$$L_m = \int_{\mathcal{P}_0} |f(\mathbf{x}, 0)| \mathrm{d}\mathbf{x} + \int_{\mathcal{P}_1} |f(\mathbf{x}, 1)| \mathrm{d}\mathbf{x} + \int_{\mathcal{P}_0^*} \left\| \phi(\mathbf{x}, 1) - \mathbf{x}^1 \right\| \mathrm{d}\mathbf{x} \ . \tag{15}$$

The last term is used to indicate a double integral over $\Omega$ and $\mathcal{I}$, which is needed as $\phi(\mathbf{x}, 1) = \mathbf{x}^0 + \int_0^1 \mathcal{V}(\phi(\mathbf{x}^0, \tau)) \mathrm{d}\tau$. We use the forward Euler step, as described in Sec. 3.2 to integrate during training. Moreover, the last term also implies that one-to-one correspondence is needed for the given two point clouds. However, thanks to the spatial smoothness of the velocity field, we only need a small part of the given correspondence to achieve satisfactory results, thus $\mathcal{P}_0^* \subset \mathcal{P}_0$ is the set of points for which correspondence information is available. We will discuss the number of correspondences that are needed in Sec. 4. Our total loss term is defined as

$$L = \lambda_f L_f + \lambda_v L_v + \lambda_m L_m \ , \tag{16}$$

where $\lambda_f$, $\lambda_v$, and $\lambda_m$ are weights to balance the joint training of velocity and implicit function term.

## 4 EXPERIMENTS

**Neural network architectures and implementation**   To ensure smooth and diffeomorphic transformations between shapes, as discussed in Sec. 3, we use the following architectures for the two neural networks: (i) Velocity-Net $\mathcal{V}$ consists of 8-layer MLP with 256 nodes per layer; (ii) Implicit-Net $f$ also consists of 8-layer MLP with 512 nodes per layer. We use Softplus Zhao et al. (2018) as the activation function. To handle high-frequency information and maintain a diffeomorphism, we incorporate a Lipschitz continuous positional encoding Yang et al. (2021).

**Datasets**   We evaluated our methods using several datasets: **Faust** Bogo et al. (2014), **SMAL** Zuffi et al. (2017), **SHREC16** Cosmo et al. (2016) and **DeformingThings4D** Li et al. (2021). Faust and SMAL provide shapes with different categories and movements. Cross-category deformations involve non-rigid transformations between distinct objects, often with significant topological changes. Movement deformation involves changes in gestures within a single object, adhering to physical laws such as rigidity or conformality. DeformingThings4D provides continuous ground-truth displacements of the source mesh vertices in each frame. We used this data set to evaluate our interpolated meshes.

**Training strategy**   To generate training data, we sample $20,000$ points on the surface of each mesh to create point clouds with partial correspondences. Each point cloud maintains ground-truth correspondences between $5\%$ to $20\%$ of its points We jointly estimate the velocity and the time-varying signed distance field. To ensure the good initialized deformation of the implicit surfaces, we train $2,000$ warm-up epochs only for velocity fields. Then we gradually increase the loss term $L_f$ weight $\lambda_f$ for Implicit-Net. We implement our code using Jax Bradbury et al. (2018) to enable fast higher-order derivative computations. We train for a total of $10,000$ epochs with batch size $4,000$. The run time is approximately 20 minutes on a GeForce GTX TITAN X GPU with CUDA for each pair.

### 4.1 SHAPE DEFORMATION

We test our method on various deformation scenarios and compare it with other methods: LipMLP from Liu et al. (2022) uses MLP layers that satisfy Lipschitz continuity to ensure smooth transitions between source and target shapes. NFGP from Yang et al. (2021) deforms shapes based on a source mesh and user-defined handle points. They estimate the implicit neural surface of the source mesh and then compute the deformed surface to match the target handle points. That means the method requires separate training for each time step. NISE fits two neural networks to the source and target meshes and trains an implicit field with a time dimension to estimate intermediate deformations. We use ground truth meshes to train both NISE and NFGP. NISE takes about 1.5 hours, and NFGP takes around 15 hours for each deformation step (over 75 hours for five steps). Our method requires only $1/5$ of the time compared to NISE, excluding the pre-training time for SDF networks of the source and target meshes, which takes around another 20 minutes.

**Extrinsic (pose) deformation** Extrinsic deformation refers to only pose changes. The transformations occurring within the same object category and no changes to the object type. In this context, we incorporate a divergence-free constraint (c.f. Eq. (7) with $\lambda_{\text{div}} > 0$) to ensure that only the object volume does not change during the deformation. Fig. 3 illustrates the results on the Faust dataset, benchmarked against other methods. While other methods fail to produce physically plausible intermediate shapes, both NFGP Yang et al. (2021) and our method successfully recover reasonable shapes. However, NFGP requires user-defined handle points for each step and must be trained incrementally, preventing it from generating a smooth deformed implicit neural surface.

**Instrinsic (non-rigid) deformation** Non-rigid deformation typically involves different objects for the source and target point clouds. In this case, we disable the divergence-free constraint by setting $\lambda_{\text{div}} = 0$. Fig. 4 shows an example that includes different categories and poses deformation (intrinsic and extrinsic) in the source and target point clouds. Since NFGP Yang et al. (2021) cannot handle non-rigid deformation, we only provide qualitative visualization results compared with the other two methods. While all methods can recover the source and target meshes, the comparison methods fail to generate reasonable intermediate meshes.

**Quantitative evaluation** To quantitatively evaluate the interpolated meshes, we use the fox and bear animation from the DeformingThings4D Li et al. (2021) dataset. These examples contain relatively small deformations per frame, making them suitable as ground truth baseline. Each sequence contains 55 meshes. For each dataset, we select 5 key meshes and estimate the deformations between them. We calculate the Chamfer Distance (CD) and Hausdorff Distance (HD) of the recovered meshes compared to the ground truth. We compare our method with LipMLP Liu et al. (2022) and NISE Novello et al. (2023). Fig. 5 shows the average error table over the 55 interpolated meshes (left) and the error plot for each mesh in the bear dataset (right). While all methods accurately recover the mesh at input time steps, LipMLP and NISE exhibit increasing errors at intermediate steps. Our method maintains a consistently low error on the intermediate meshes. Visualization results and error plots for the fox dataset are provided in Appendix A.

## 4.2 INCOMPLETE AND SPARSE INPUT

Implicit methods showcase remarkable flexibility in representing shapes with varying typologies. In this section, we show some challenging cases that our method can still tackle.

**Different sparsity inputs** Most existing approaches require fitting two separate networks to estimate the Signed Distance Fields (SDF) for the initial and final shapes, as highlighted in previous works Novello et al. (2023); Yang et al. (2021); Liu et al. (2022). Consequently, the quality of the results is heavily dependent on the characteristics—such as the sparsity of the source and target point clouds. If the fitting process for one shape is unsuccessful, these methods fail to estimate intermediate shapes. Our approach overcomes these challenges through the Velocity-Net, which enables tracking the dense initial point cloud $\mathcal{P}_0$ to the sparse final point cloud $\mathcal{P}_1$, and recovering the underlying shapes without compromising result quality. Fig. 7 illustrates the varying sparsity levels of the input data. The source point cloud $\mathcal{P}_0$ contains $\sim 20,000$, points while the target point cloud $\mathcal{P}_1$ only has $\sim 2,000$ points. While the source point cloud is dense enough to train a neural network for fitting an SDF, the target point cloud is much sparser than the source. Although one could employ densification

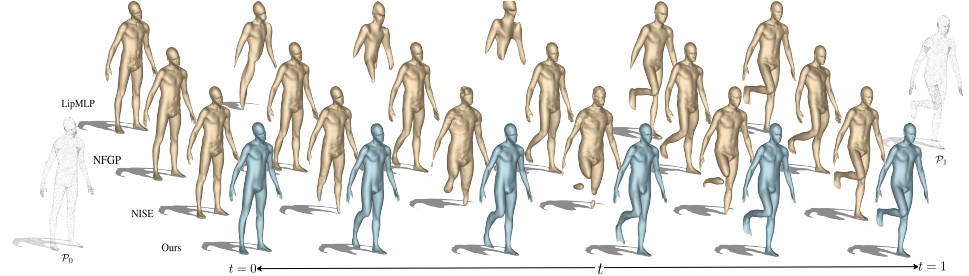

Figure 3: Experiment on extrinsic deformation. LipMLP Liu et al. (2022) and NISE Novello et al. (2023) fail to estimate the physically plausible intermediate shapes. NFGP Yang et al. (2021) can recover reasonable meshes but it is trained separately for each time step. Our method can recover realistic intermediate shapes in one model.

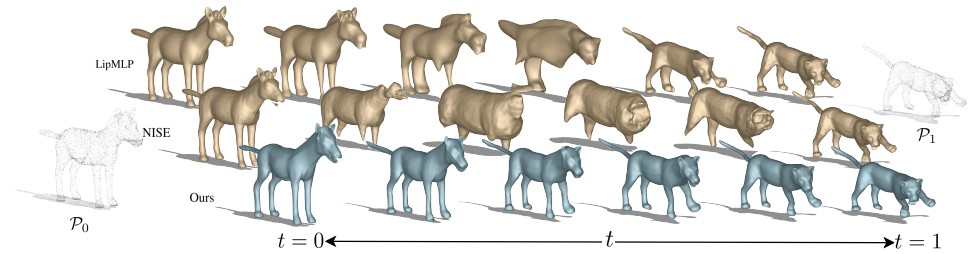

Figure 4: Experiment involves both extrinsic and intrinsic deformation. While LipMLP( Liu et al. (2022)) and NISE( Novello et al. (2023)) fail to create reasonable middle-step meshes, our method generates appropriate transition meshes from two given point clouds.

| method | metric | datasets | |
|--------|--------|----------|------|
| | | fox | bear |
| LipMLP | CD↓ | 1.745 | 2.649 |
| | HD↓ | 2.456 | 1.401 |
| NISE | CD↓ | 0.178 | 0.366 |
| | HD↓ | 0.262 | 0.560 |
| Ours | CD↓ | **0.108** | **0.260** |
| | HD↓ | **0.114** | **0.265** |

(a) Average error evaluated on intermediate meshes. LipMLP Liu et al. (2022) and NISE Novello et al. (2023) can fit well for the given meshes but produce big errors at the intermediate meshes. Our proposed method maintains a small error even on the middle shapes.

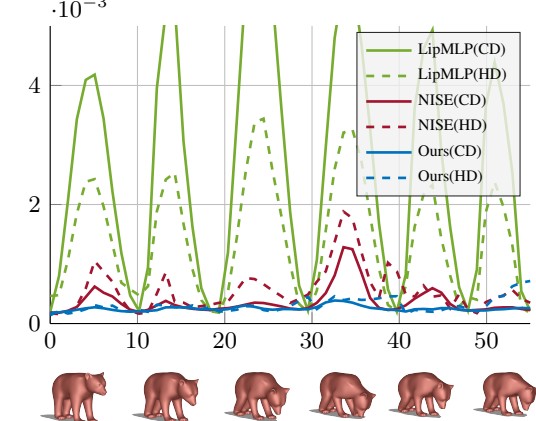

(b) Error plot of each intermediate mesh of bear dataset.

Figure 5: Quantitative evaluation of the deformed shapes. Chamfer Distance (CD) scaled by $10^3$ and Hausdorff Distance (HD) scaled by $10^2$ for the 55 intermediate shapes.

strategies Sang et al. (2023); Zhao et al. (2022) or utilize priors Ma et al. (2022) before fitting the networks, our method successfully reconstructs both the final and intermediate shapes without any additional modifications.

**Incomplete inputs** Additionally, our time-varying implicit network is capable of completing the shape even when both input sets are incomplete. We study the situation that the given point clouds are incomplete in different areas. We sample point clouds from the incomplete shapes $\mathcal{S}_0$ and $\mathcal{S}_1$ to create point clouds with holes $\mathcal{P}_0$ and $\mathcal{P}_1$ as input data. Results in Fig. 6 demonstrate that our method can complete the shape. This experiment re-

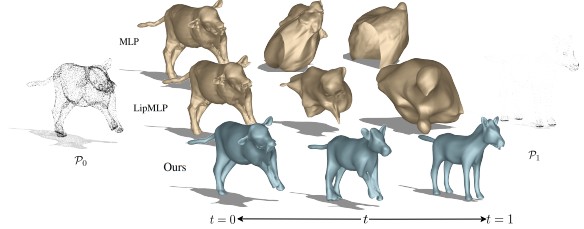

Figure 7: Compared to previous methods, our method can still recover both the target shape and the intermediate shapes even one input point cloud is excessively sparse.

sult implies that we do not need consistent topology from the inputs, i.e. we can handle inputs with different topology features. Note that this is typically challenging for mesh-based methods, both with and without ground truth correspondences. Meshes have a fixed topology and deforming a mesh to another one with substantially varying topology is a challenging endeavour. Moreover, mesh-based methods struggle with completing the mesh without giving a complete shape as a prior. We provide an analysis and comparison with mesh-based methods in Appendix A.

### 4.3 ABLATIONS

**Modified level-set equation** In this section, we demonstrate that the modified level-set equation Eq. (13) leads to more stable results compared to the original level-set equation. Fig. 8 shows the

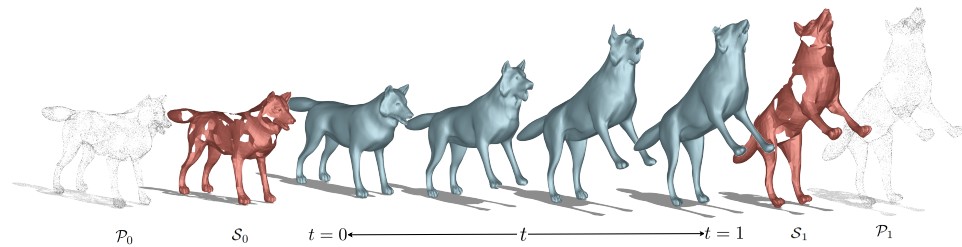

Figure 6: We sample point cloud $\mathcal{P}_0$ and $\mathcal{P}_1$ from ground truth meshes with holes $\mathcal{S}_0$ and $\mathcal{S}_1$, respectively. The input point clouds are incomplete. The proposed method can still recover neural implicit surfaces with complete shapes together with the intermediate steps.

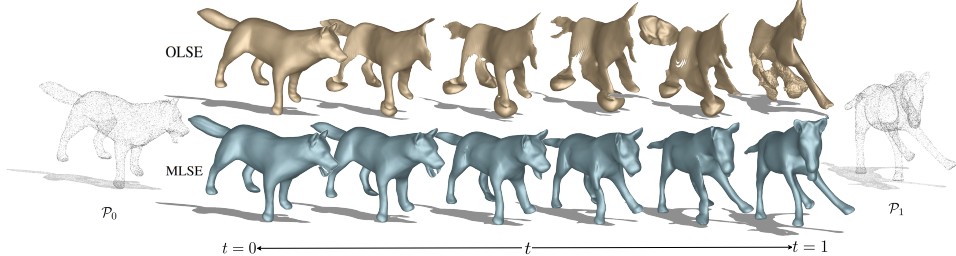

Figure 8: Ablation study for original level-set equation (OLSE) Eq. (10) (first row) and modified level set equation (MLSE) Eq. (13) (second row). The OLSE does not coincide with the eikonal constraint Eq. (3) while MLSE implies it. The reconstruction results show the MLSE produces a more stable and non-degenerate signed distance field.

qualitative results of the two different level-set equations. The Original Level-Set Equation (OLSE) uses the formulation Eq. (10) plus Eq. (3) at every discrete time step in training. The Modified Level-Set Equation (MLSE) embeds the Eikonal constraint compactly. Fig. 8 shows that MLSE prevents the degenerated meshes.

**Volume preservation effect** We explore the influence of our divergence-free regularizer Eq. (7) proposed in Sec. 3.2. We show that the divergence-free regularizer on the velocity field indeed preserves the total volume of the recovered shape. Fig. 9a shows that if the divergence-free constraint is enforced, the network produces a surface that preserves the volume of the source point cloud. It generates meshes that adopt the appearance features of the target point cloud but do not expand the volume to fit the target point cloud. The whole visualization is provided in Appendix A.

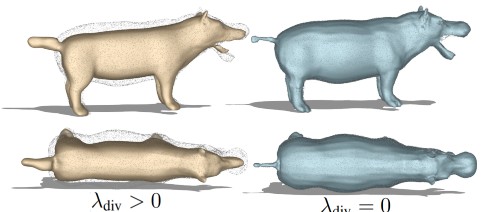

| dataset | Metric | 1% | 5% | 10% | 20% |
|---------|--------|------|------|------|------|
| fox | CD↓ | 0.119 | 0.108 | 0.111 | 0.110 |
| | HD↓ | 0.148 | 0.141 | 0.114 | 0.106 |
| bear | CD↓ | 0.257 | 0.250 | 0.251 | 0.250 |
| | HD↓ | 0.310 | 0.306 | 0.308 | 0.291 |

(a) Visualization of the output meshes with and without divergence-free loss. From the same source point cloud, the left side of the mesh is slim compared to the target point cloud while the right side mesh can fit perfectly.

(b) Error with different sparsity levels. The reconstruction accuracy increases along with the number of GT correspondences. However, even with 1% ground truth correspondence, our method can get small CD and HD errors.

Figure 9: Divergence-free constrains ablation (left) and Sparsity correspondence ablation (right).

**Sparsity of the correspondences** Our method utilizes a certain amount of ground truth correspondence. In this section, we explore the influence of correspondence numbers on the deformation qualities. We sample correspondences in different proportions to the point cloud numbers: 1%, 5%, 10%, 20% to test the recovered intermediate mesh quality.

**Noisy correspondences ablation** We test the robustness of our method in different ways. First, we test against local noise on ground-truth correspondences. The test data contains 5% correspondences with respect to the total number of input points. We sample 5%, 10%, and 20% of the correspondences and swap them with their 5th nearest neighbor correspondences (see Fig. 10). We also test against

the global noise, where we randomly swap them with other correspondences, regardless of whether the swapped correspondences are neighboring. This represents an extreme case for noise simulation. Due to page limitation, we only show the local cases in the main paper. For more ablations please refer to the appendix.

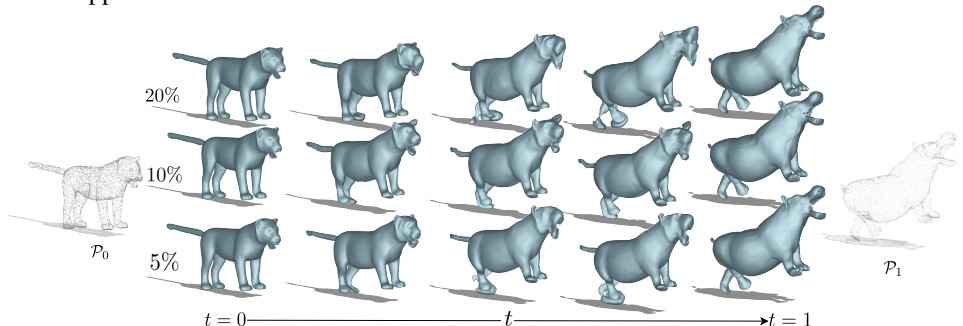

Figure 10: We add noise to the correspondences by randomly choosing different portions of the GT correspondences and swap them with its $5th$ nearest neighbor correspondences. Qualitative results show that our method is stable up to $10\%$ correspondences and still gives relatively reasonable results up to $20\%$ misaligned correspondences.

**Combining other methods to obtain correspondences** Our proposed method integrates seamlessly with existing point-registration techniques when ground-truth correspondences are unavailable. In this section, we present results using a prior method Cao et al. (2024a) to first obtain correspondence vertices, followed by the application of our method. Many point-registration approaches provide only sparse correspondence pairs with small deviations. However, due to the robustness of our method and the fact that we require only around $10\%$ of correspondences to achieve strong results, we can effectively utilize most of their output. Fig. 11 illustrates the results of our method built on top of a matching technique. For more results using obtained correspondences, please refer to Appendix A.

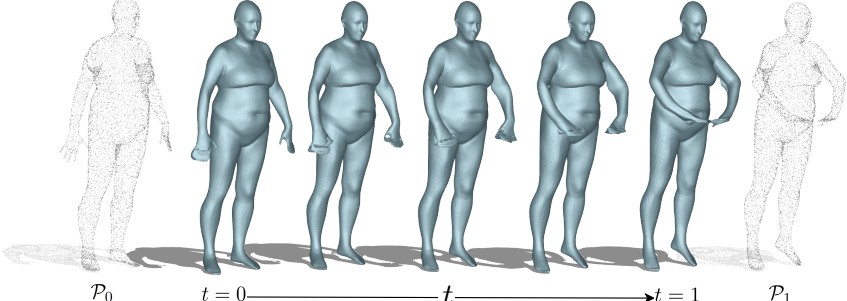

Figure 11: Results using correspondences obtained by other methods Cao et al. (2024a). The method offers 5000 correspondences to each pair of the shapes.

## 5 DISCUSSION

**Limitations and future works** Even though our work is self-supervised, we still need sparse correspondences of the point clouds. Moreover, due to the lack of neighboring information, our method struggles with large deformations and may introduce artifacts around the reconstructed surfaces (c.f. Appendix A Fig. 23). In the future, we will explore dealing with large deformations and extend our work to dynamic implicit surface generation.

**Conclusion** In this paper, we introduce a method to recover the implicit surface of two given point clouds based on sparse correspondences, while also generating a natural and physically plausible intermediate deformation. Our approach does not require any intermediate shape supervision beyond the provided source and target point clouds. Our method directly deforms the implicit field using an explicitly estimated velocity field, enabling us to estimate deformations directly from the point cloud input. This approach allows for the representation of more flexible topologies and can handle more challenging scenarios. Our method also broadens the application of implicit representations from static objects to dynamic objects.

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

# A  APPENDIX

## A.1  MATHEMATICAL PROOFS

**Existence of the velocity field**    We assume that our velocity field is the solution of Eq. (4). The existence of the solution for the differential equation is given by Picard-Lindelöf Theorem Arnold (1992) which states that for $\phi : \mathbb{R} \times \mathbb{R}^n \to \mathbb{R}^n$, for continuous $t$ and Lipschitz continuous $\mathcal{V}$, then there exist an $\epsilon > 0$ such that the initial value problem Eq. (4) have unique solution $\mathcal{V}$ on interval $[t - \epsilon, t + \epsilon]$. As we choose $\mathcal{V}$ smooth enough, the requirement is satisfied.

**Velocity fields that generate diffeomorphism**    Dupuis et al. (1998a) proved the existence of the smooth trajectory generated by Eq. (4) depends on the smoothness constraint in the vector field $V$. They also proved that choosing $V$ such that $V$ is a smooth and compactly-supported vector field with an inner product defined by a differential operator $\mathcal{L}$ ensures the solution in the space of diffeomorphism.

**Smooth operator $\mathcal{L}$**    The differentiable operator $\mathcal{L}$ introduced in Eq. (6) is chosen to have the type $\mathcal{L} = -\alpha \Delta + \gamma \mathbf{I}$, where $\alpha$ enforces the smoothness and $\gamma > 0$ ensures the operator is non-singular. In the experiments, we set $\alpha = 0.01$ and $\gamma = 1$. The velocity field is a Hilbert space defined by the operator $\mathcal{L}$ with norm

$$\|\mathcal{V}\|_V = \langle \mathcal{V}, \mathcal{LV} \rangle . \tag{17}$$

Beg et al. (2005) proved that this type of choice for operator $\mathcal{L}$ stratifies the requirement that $\phi$ is a diffeomorphism Dupuis et al. (1998b).

**Level-set equation**    Following Eq. (9) and smoothness assumption of $\Omega$ and $f$ we have

$$\int_\Omega \frac{\mathrm{d}f(\phi(\mathbf{x}, t))}{\mathrm{d}t} \mathrm{d}\mathbf{x} = 0 , \tag{18}$$

we have $\frac{\mathrm{d}f(\phi(\mathbf{x},t))}{\mathrm{d}t} = 0$ almost everywhere. Compute the derivatives, we have

$$\frac{\mathrm{d}f(\phi(\mathbf{x}, t))}{\mathrm{d}t} = \partial f_t + \partial f_\mathbf{x} \partial \phi_t = 0 . \tag{19}$$

Together with Eq. (4) and $\partial_\mathbf{x} f = \nabla f$, we have

$$\frac{\mathrm{d}f(\phi(\mathbf{x}, t))}{\mathrm{d}t} = \partial f_t + \nabla f \cdot \mathcal{V} = 0 , \tag{20}$$

which is the original level-set equation.

**Modified Level-set equation**    To ensure the Eikonal constraint on continuous time-space for any $t$, it is equivalent to solving an additional initial problem of the PDE, i.e.

$$\begin{cases} \frac{\mathrm{d}}{\mathrm{d}t} \|\nabla f(\mathbf{x}, t)\| = 0, & t \in \mathcal{I} , \\ \|\nabla f(\mathbf{x}, 0)\| = 1 . \end{cases} \tag{21}$$

The function above ensures that $\|\nabla f(\mathbf{x}, t)\| = 1$ for any $t \in \mathcal{I}$. Eq. (13) holds on the zero-crossing surface $\partial \Omega$ because the function value $f$ is zero at the zero-cross surface and the two term $\partial f_t + \mathcal{V} \cdot \nabla f$ and $\mathcal{R}$ both equal zero in the surface domain $\Omega$. It is more compact to solve Eq. (13) than solve Eq. (10) plus Eq. (21) separately since the later solves two PDEs Bogo et al. (2014).

Note that Eq. (13) needs two initial value $f(\mathbf{x}, 0) = f^0$ and $\|\nabla f(\mathbf{x}, 0)\| = 1$. We explain in the Appendix A.2 how we avoid pre-training a network to satisfy the initial condition.

## A.2  TRAINING STRATEGY

We implement our method using Jax Bradbury et al. (2018) and set the learning rate to 0.005 with a decay rate 0.5 within interval 2000. We initialize Implicit-Net's weights and bias such that it represents a sphere at step 0, following the method proposed in Gropp et al. (2020).

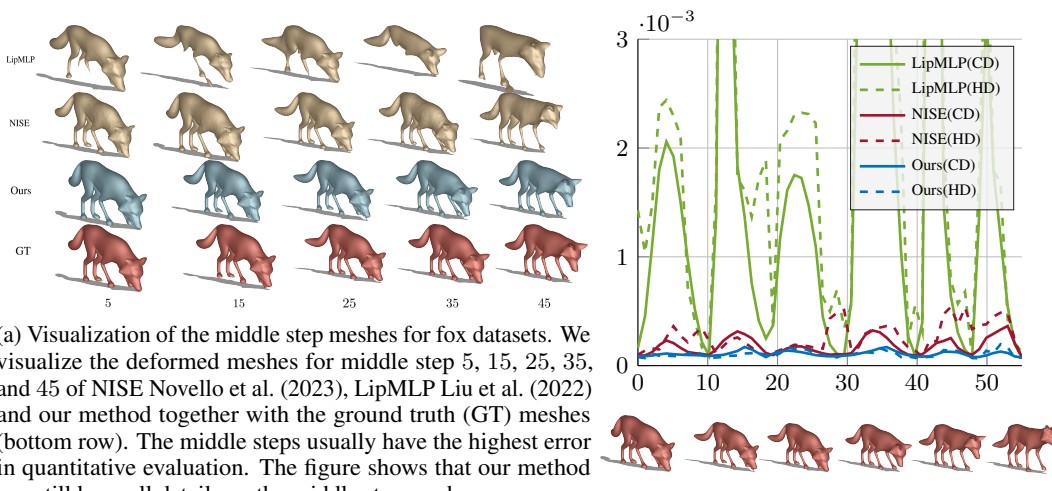

(a) Visualization of the middle step meshes for fox datasets. We visualize the deformed meshes for middle step 5, 15, 25, 35, and 45 of NISE Novello et al. (2023), LipMLP Liu et al. (2022) and our method together with the ground truth (GT) meshes (bottom row). The middle steps usually have the highest error in quantitative evaluation. The figure shows that our method can still keep all details on the middle step meshes.

(b) Error plot of the fox dataset.

Figure 12: Quantitative and qualitative evaluation of the deformed shapes. Chamfer Distance (CD) scaled by $10^3$ and Hausdorff Distance (HD) scaled by $10^2$ for the 55 intermediate shapes.

**Invertible Lipschitz Positional Encoding**  We adopt the invertible Lipschitz positional encoding same as Yang et al. (2021) to cooperate with MLP in both velocity net and implicit net to produce a stable output.

$$\gamma_i(\mathbf{x}) = \frac{1}{\sqrt{2m+1}}(x_i, \frac{\cos(2^0\pi x_i)}{2^0\pi}, \frac{\sin(2^0\pi x_i)}{2^0\pi}, \ldots, \frac{\cos(2^m\pi x_i)}{2^m\pi}, \frac{\sin(2^m\pi x_i)}{2^m\pi}) . \tag{22}$$

**Oriented point cloud**  Our method does not require an oriented point cloud (point cloud with normal). However, if the normal information $\{\mathbf{n}\}^i$, for $i = \{0, 1\}$ is available for the given point clouds, the second-order constraints on matching loss can be added according to Eq. (2). The normal loss term is

$$L_n = \int_{\mathcal{P}_i} |1 - \langle \frac{\nabla f}{\|\nabla f\|}, \mathbf{n}^i \rangle| d\mathbf{x}, \text{ for } i \in \{0, 1\} . \tag{23}$$

$L_n$ term can accelerate the convergence speed of Implicit-Net at time $t = 0$ and 1.

**Network initialization**  We initialize Implicit-Net such that it represents a unit sphere at time 0 Gropp et al. (2020). Thus it is a valid signed distance field that satisfies the initial condition in Eq. (21). To satisfy the initial condition in Eq. (13) without pre-train the net on $f(\mathbf{x}, 0)$, we set $\lambda_m$ much larger than $\lambda_f$ such that the network first converges at time 0 to fit $f(\mathbf{x}, 0) = 0$ on the given input point cloud $\mathcal{P}_0$. Thus, for the experiment showed on the paper, we set $\lambda_f = 100$, $\lambda_m = 200$, $\lambda_v = 20$ and $\lambda_l = 10$.

**Warm up training**  As described in Sec. 4, we first freeze the implicit network, i.e. we set $\lambda_f = 0$, $\lambda_v = 20$, $\lambda_m = 100$ for first 2000 epochs, then we gradually increase the it using $\lambda_f = \frac{k-2000}{n-2000}100$ for $k < n$, and $\lambda_f = 100$ for $k \geq n$, where $k$ is the $k$th-epoch, and $n = 5000$. As we observe that velocity field convergence is faster than the implicit field, we decrease the velocity loss to train only implicit net after a certain epoch, i.e. $\lambda_v = 0$ for $k > 8000$.

### A.3  QUANTITATIVE RESULTS

We show the error plot for fox datasets in which the average error number is reported in Fig. 5a. Even with small deformation between each key mesh, the comparison methods still report high errors on the middle step meshes. We show the ground truth meshes with the middle steps in Fig. 12a and the error plot for each mesh in Fig. 12b.

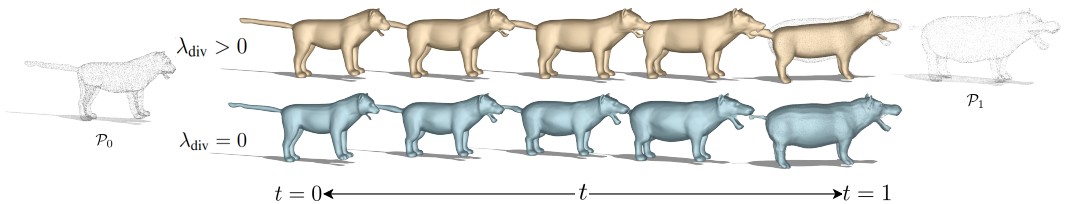

Figure 13: Full visualization of Fig. 9a in main paper.

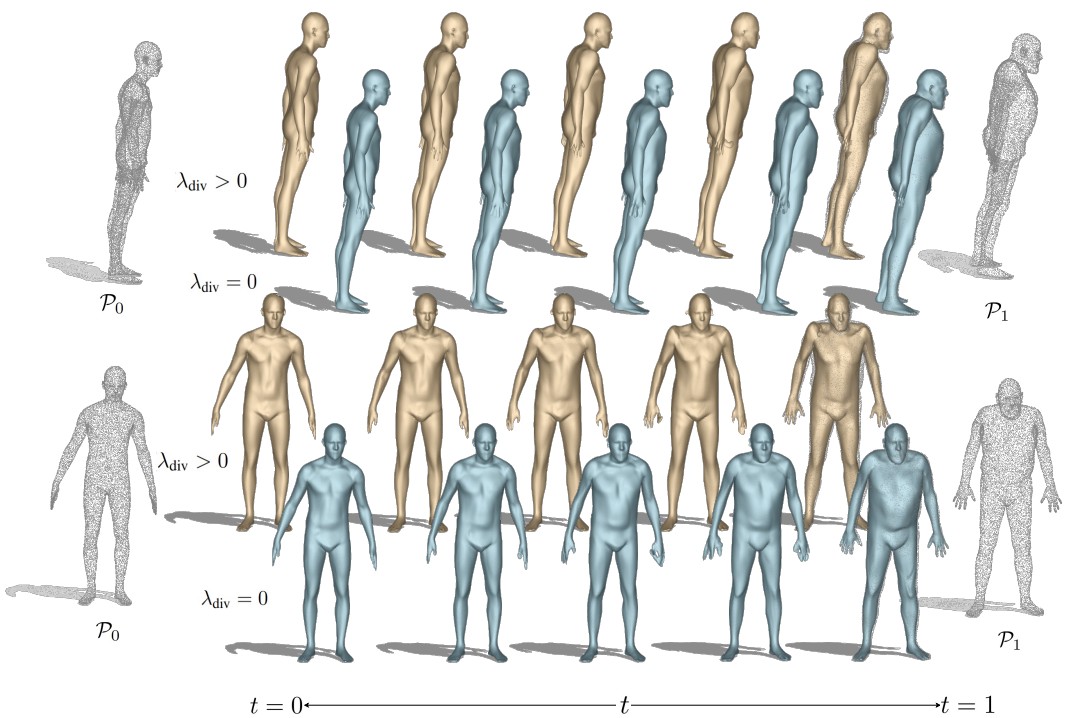

Figure 14: Visualization of divergence-free constraint on Faust dataset. With $\lambda_{\text{div}} > 0$ (yellow meshes), the deformed shapes are still slim and only adopt the movement of the target point cloud $\mathcal{P}_1$, while $\lambda_{\text{div}} = 0$ (blue meshes) the deformed meshes have the same gesture and body shape of the target point cloud $\mathcal{P}_1$

.

## A.4 DIVERGENCE-FREE CONSTRAINT ABLATION

In this section, we visualize the deformed meshes under two different settings: with divergence-free term ($\lambda_{\text{div}} > 0$) and without divergence-free term ($\lambda_{\text{div}} = 0$). In Fig. 13, the recovered deformation meshes stay slim and thin when $\lambda_{\text{div}} > 0$ (top row) and only adopted features such as the shape of the mouth of the target point cloud. When $\lambda_{\text{div}} = 0$ (bottom row), the deformed meshes can perfectly fit the target point cloud, which means the volume expanded compared to the source point cloud. Fig. 14 shows another example of the volume persevering effect.

## A.5 LAPLACIAN CONSTRAINT ABLATION

In this section, we show the visual ablation of Laplacian constraint equation 6. Our smoothness ablation on the velocity field ensures spatial smoothness over the integration domain, which is particularly helpful for very sparse correspondences.

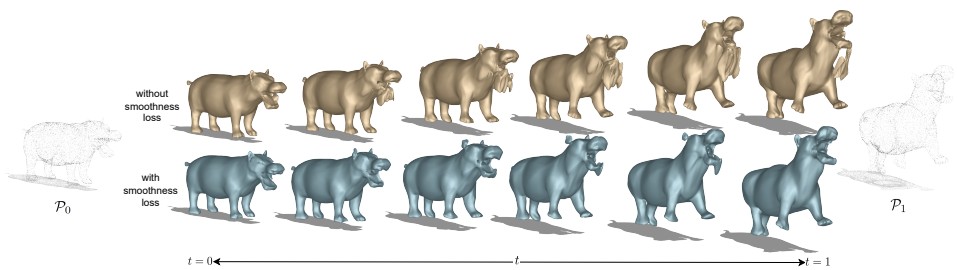

Figure 15: Our smoothness ablation on the velocity field ensures spatial smoothness over the integration domain, which is particularly helpful for very sparse correspondences.

### A.6 MODIFIED LEVEL SET EQUATION ABLATION

In this section, we show additional visualization results of our proposed modified level set equation (MLSE) with original level set equation (OLSE). Comparing MLSE to OLSE, enforcing the Eikonal loss at intermediate time steps is challenging with OLSE. This process involves moving the points using velocity and then enforcing the Eikonal constraint on the moved points, which can cause a coupling effect that leads to the degeneration of the implicit field or velocity field. As shown in Fig. 16, the first two rows illustrate that, while the final mesh fits the target, artifacts are created in the intermediate steps. The bottom two rows demonstrate a topology change in the point cloud (e.g., the crossed legs of the cat separate later). OLSE degenerates in the middle steps, and due to the continuity of the function, it retains the degenerated legs even when fitted to the target point cloud.

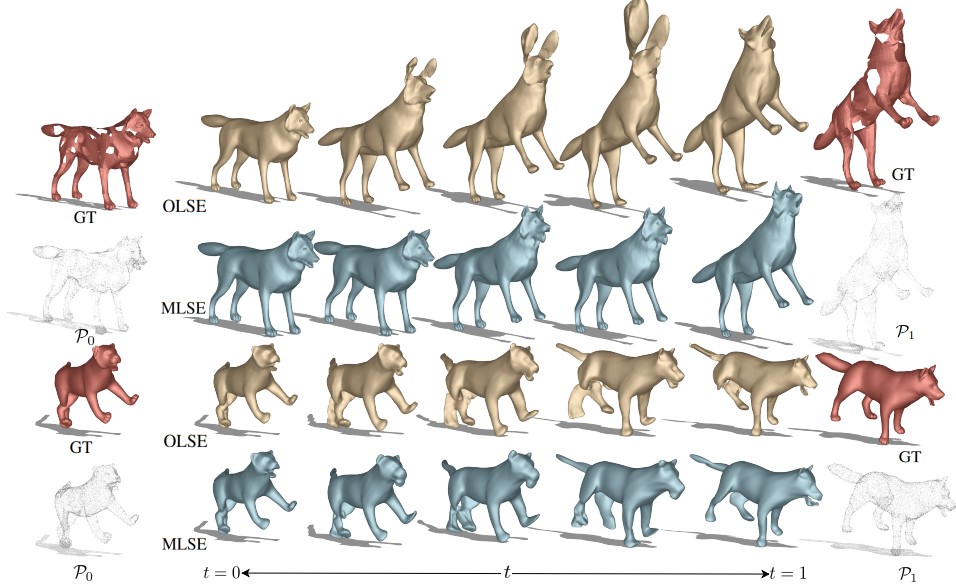

Figure 16: Additional ablation results for MLSE and OLSE.

### A.7 DETAIL PRESERVING

We show the results for meshes containing more complicated details. We deform the original Armadillo mesh using Blender to create the target shape and run our method to interpolate the intermediate shapes. As shown, our method can preserve most of the complicated geometry details.

### A.8 CORRESPONDENCE SPARSITY ANALYSIS

In this section, we present the qualitative results of our method across varying numbers of ground truth correspondences. We generated input point clouds by sampling $20,000$ points and assessed deformation quality at approximately $1\%$, $5\%$, and $10\%$ correspondence levels. As illustrated

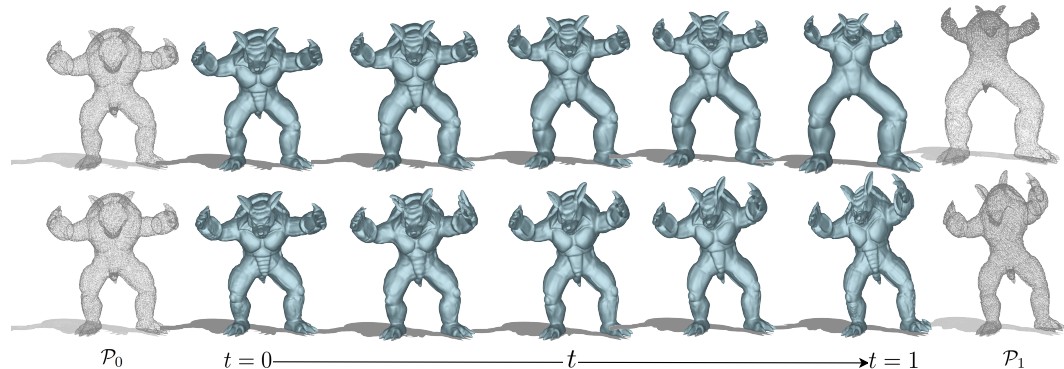

$\mathcal{P}_0$     $t = 0$————————— $t$ ———————————→$t = 1$     $\mathcal{P}_1$

Figure 17: We show extra results on Armadillo for showing the detail-persevering of our method. Our method can preserve most of the complicated geometry details while producing physically plausible intermediate shapes.

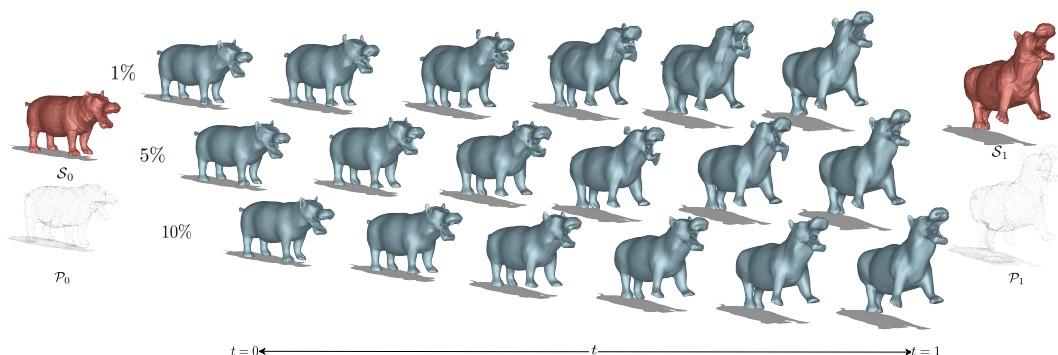

Figure 18: We explore the impact of the number of correspondences on the quality of the final deformation. When only a small percentage of correspondence is used, artifacts tend to appear in the intermediate shapes. Remarkably, our method achieves reasonable estimations with as few as approximately $1\%$ ground-truth correspondences. Furthermore, when more than approximately $10\%$ correspondences are available, our proposed method consistently delivers high-quality results.

in Fig. 18, we display the intermediate shapes produced using different quantities of correspondences during training. Here, $\mathcal{S}_0$ and $\mathcal{S}_1$ are the ground truth meshes. $\mathcal{P}_0$ and $\mathcal{P}_1$ are the sampled point cloud inputs. Our method effectively handles different sparsity levels of correspondences and delivers high-quality results when approximately $5\%$ of the correspondences are available.

### A.9 NOISY CORRESPONDENCES ANALYSIS

In this section, we show additional visualization of the local noise correspondence ablation together with global noisy analysis.

For global noise on ground-truth correspondences. The test data contains $5\%$ correspondences relative to the total number of input points. We sample $1\%$, $5\%$, and $10\%$ of the correspondences and randomly swap them with other correspondences, regardless of whether the swapped correspondences are neighboring. This represents an extreme case for noise simulation. In this scenario, as shown in Fig. 19, our method produces satisfactory results with $5\%$ wrong correspondences and still gives reasonable deformation with $10\%$ misaligned correspondences.

Additionally, we show one more results on local noise ablations. The test data contains $1\%$, $5\%$ correspondences with respect to the total number of input points. We sample $5\%$, $10\%$, and $20\%$ of the correspondences and swap them with their 5th nearest neighbor correspondences (see Fig. 20).

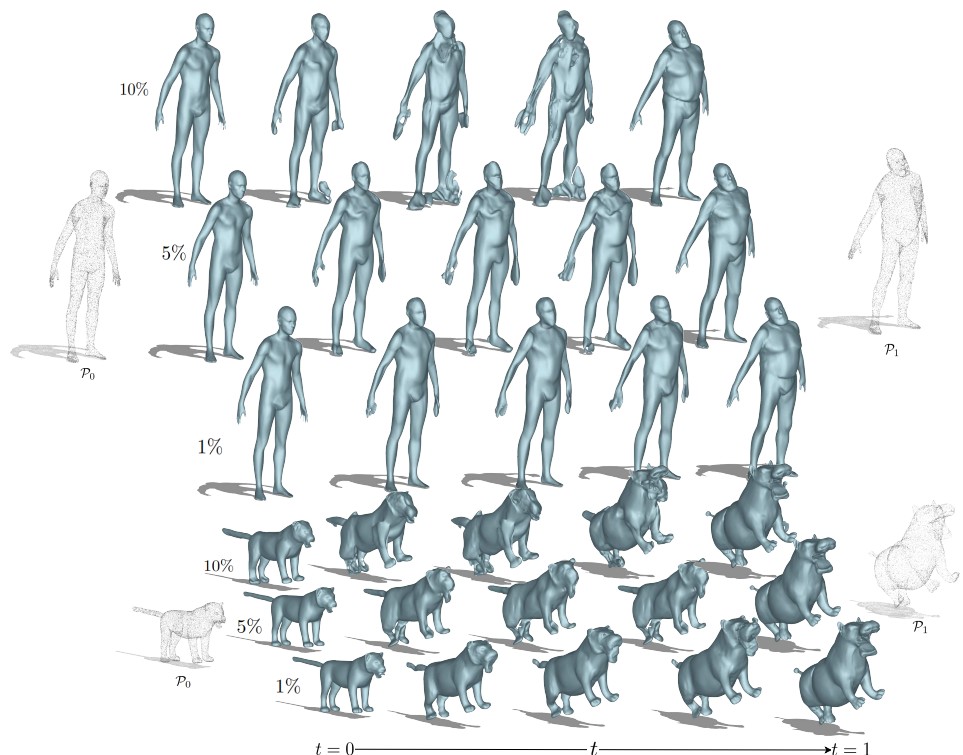

Figure 19: The input data contain around $5\%$ ground-truth correspondences. We add noise to the correspondences by randomly swapping $1\%$, $5\%$, $10\%$ of the correspondences **globally**. Qualitative results show that our method is stable up to $5\%$ error and still gives relatively reasonable results up to $10\%$ error. Note that this is an extreme situation as mismatching happens globally.

## A.10 COMPARE WITH MESH-BASED METHODS

Mesh-based surface deformation is a well-explored area, many papers have done mesh deformation with Eisenberger & Cremers (2020); Alexa et al. (2023); Vyas et al. (2021) or without correspondence Eisenberger et al. (2021); Cao et al. (2024b;a). In contrast to implicit-based methods, mesh-based methods enable more stable, artifacts-free results as no surface fitting or estimation is needed. We compare our method against the state-of-the-art mesh method, SmS Cao et al. (2024a). SmS does not require ground truth correspondences and is capable of producing physically plausible intermediate shapes. Our method achieves results comparable to SmS. As illustrated in Fig. 21. However, while our approach successfully preserves all the fine details, it tends to create artifacts around the surfaces and may result in less physically accurate deformations when the deformation is too large, we show some failure cases in Fig. 23. Moreover, our method reconstructs smoother meshes compared to SmS and GT meshes, because our implicit representation allows us to render higher-resolution meshes. On the contrary, mesh-based methods, like SmS keep the original resolution (same vertices, triangles, and faces) of input meshes.

However, in some situations, such as inconsistent topology or incomplete shape without ground truth complete shape, our method can handle these challenging scenarios. Mesh-based methods struggle in these situations. Fig. 22 shows some challenging cases. In the cat (top row) example, the source and the target point cloud are sampled from meshes that have holes in the meshes. The source and target meshes are incomplete in different areas. The second example centaur shows that case of complete source mesh but incomplete target mesh. The third example even though the source and target meshes are complete, because of the overlapped feet, and overlapped arm in the target shape, the topology of the meshes is different. These three challenging examples are not feasible for the mesh-based methods. The mesh-based methods cannot handle them because the vertices and faces are not one-to-one matches anymore, even with ground truth correspondences. Since the topology of the deformed mesh is fixed, it is not trivial to deform to a shape that has a different topology. Our

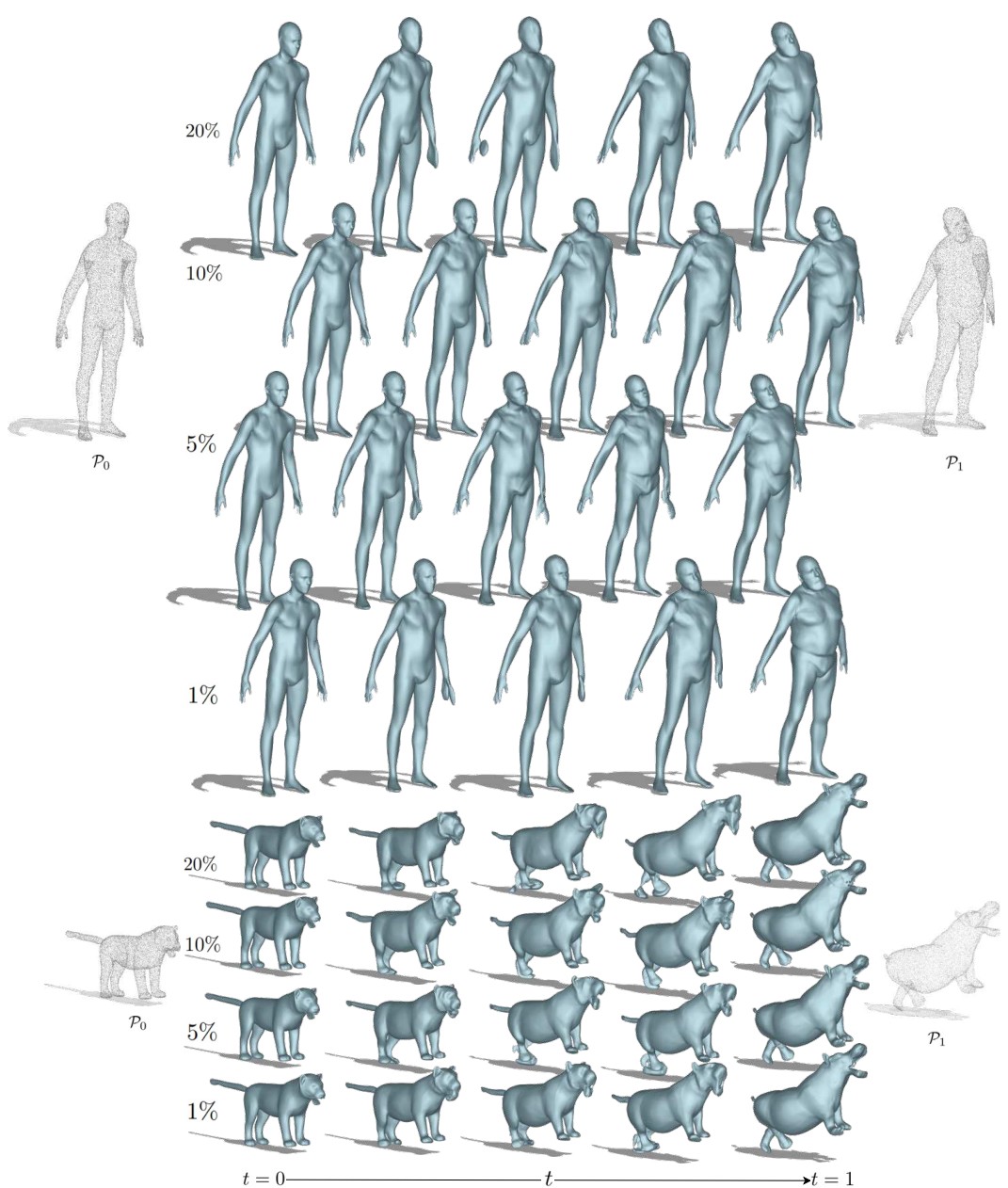

Figure 20: The input data contain around $5\%$ ground-truth correspondences. We add noise to the correspondences by randomly choosing $1\%$, $5\%$, $10\%$, $20\%$ of the GT correspondences and swap them with its $5th$ nearest neighbor correspondences. Qualitative results show that our method is stable up to $10\%$ correspondences and still gives relatively reasonable results up to $20\%$ misaligned correspondences.

method, on the other hand, uses implicit representation and does not define mesh topology explicitly. Thus, our method can handle these cases.

## A.11 FAILURE CASES

As mentioned in Sec. 5, our work has some limitations. Here, we present some failure cases and discuss potential future improvements. Compared to mesh-based methods, our approach struggles with large deformations. Fig. 23 illustrates a scenario where mesh-based methods succeed, but our

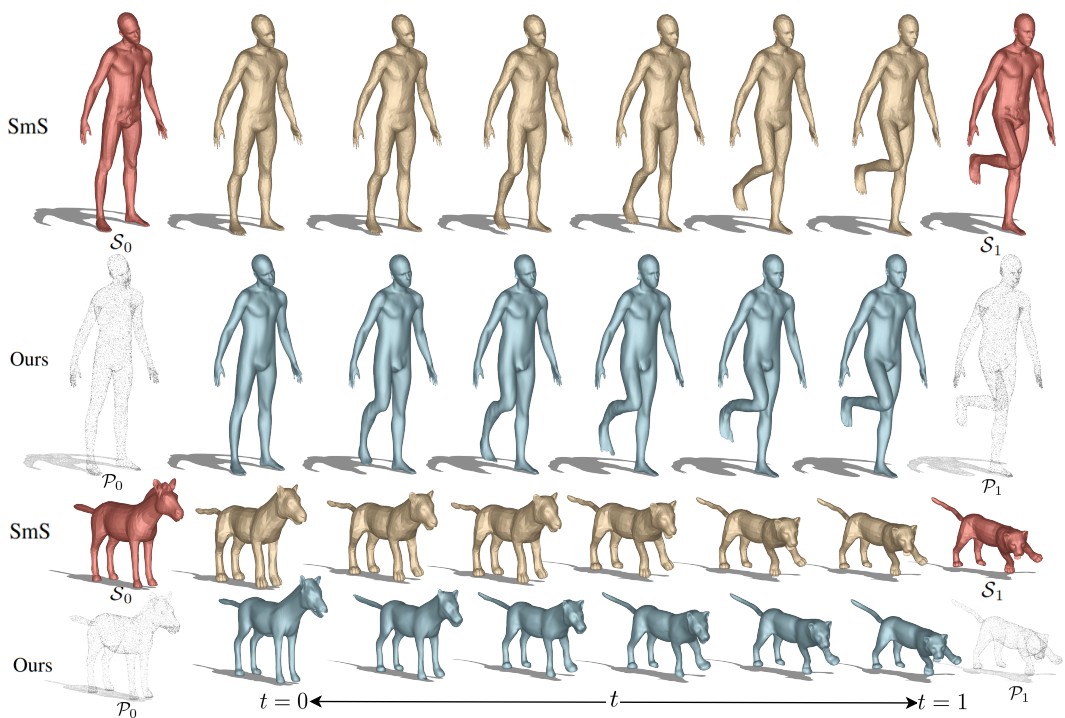

Figure 21: Comparison results against state-of-the-art mesh-based method SmS Cao et al. (2024a). We produce comparable results with the meshed-based methods. However, mesh-based methods preserve more details such as human fingers. Implicit methods, on the other hand, enable rendering arbitrary resolution meshes.

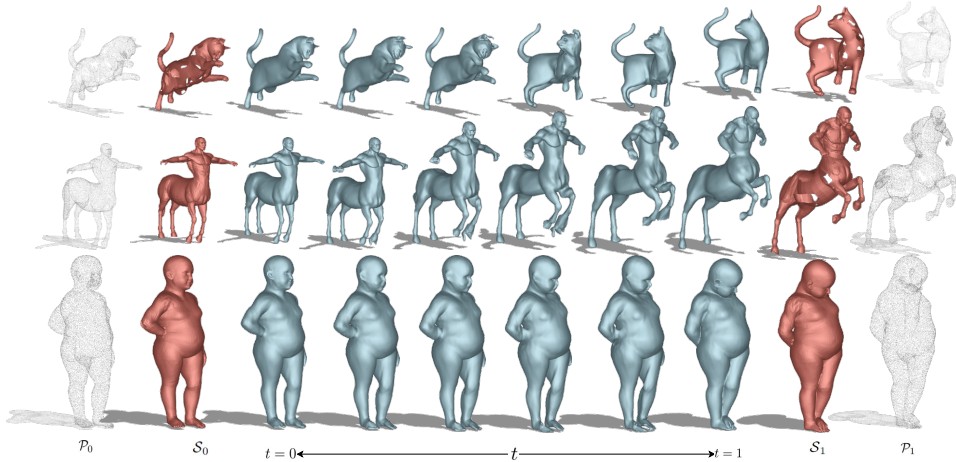

Figure 22: The proposed method can deal with inconsistent topology input, such as different incomplete shapes or self-intersect shapes.

method produces unsatisfactory results (top two rows). In the second scenario (bottom row), although mesh-based methods fail, our method also produces artifacts around the feet due to insufficient local constraints in those areas. Another limitation occurs when there are large missing areas in the source or target point clouds. Unlike smaller holes (as shown in Fig. 6 and Fig. 22), substantial missing parts result in failure cases because our Velocity-Net cannot correctly move the points to the appropriate locations.

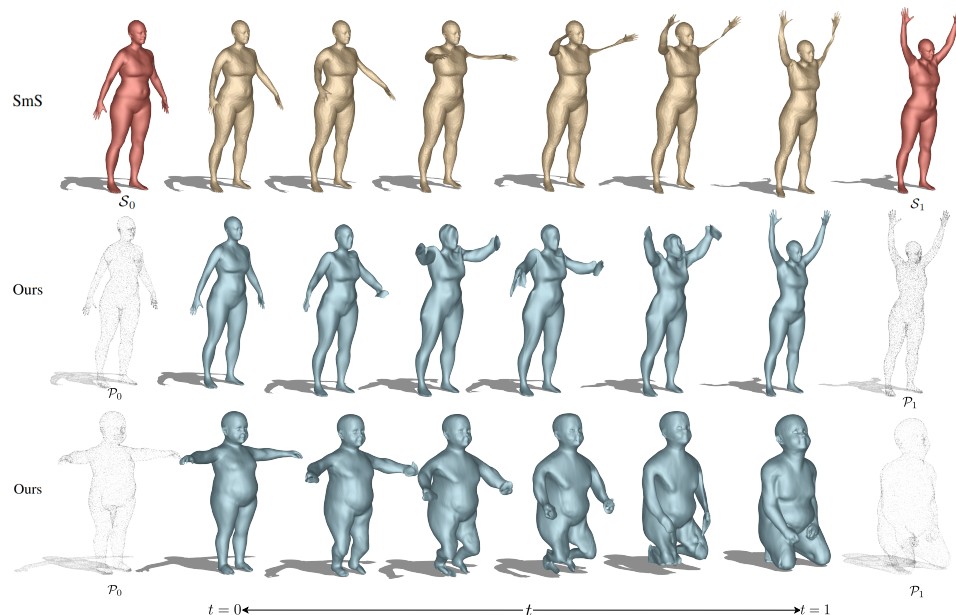

Figure 23: Failure cases of the proposed method. When the deformation is too large, our method tends to create artifacts on the surfaces.

|  | CD $(\times 10^4)$ ↓ | HD $(\times 10^2)$ ↓ | SA$\sigma(\times 10)$ ↓ | P-RMSE $(\times 10)$ ↓ |
|---|---|---|---|---|
| NFGP Yang et al. (2021) | 0.272 | **0.025** | 0.075 | ✗ |
| LipMLP Liu et al. (2022) | 14.99 | 2.125 | 1.252 | ✗ |
| NISE Novello et al. (2023) | 6.588 | 2.167 | 0.321 | ✗ |
| Ours | **0.270** | 0.047 | **0.023** | 0.024 |

Table 1: We evaluate our method with comparison methods on the **4d-Dress** Wang et al. (2024) dataset where the intermediate shapes are given. We report the average Chamfer Distance (CD) and Hausdorff Distance (HD) over 5 interpolated meshes. We also report the standard deviation of the surface area SA$\sigma$ to indicate the changes in mesh-area over the deformation. Moreover, to evaluate our velocity field, we compute the per-point Euclidean distance as Root Mean Square Error (P-MSE) over the ground truth meshes. For the comparison methods that cannot compute the per-point distance error, we mark it as ✗.

## A.12 PHYSICAL PLAUSIBLE QUANTITATIVE RESULTS

To quantitatively evaluate our method, we use the **4D-Dress** dataset Wang et al. (2024), which comprises high-frequency meshes capturing human movements. For evaluation, we select a mesh pair with four intermediate meshes as source and target inputs. The four intermediate meshes serve as ground truth. We compute the Chamfer Distance (CD) and Hausdorff Distance (HD) to quantitatively assess the accuracy of our method. Additionally, to demonstrate that our approach preserves volume, we report the standard deviation of the surface area across all interpolated meshes, verifying that the meshes do not overstretch or shrink during deformation. The results are summarized in Tab. 1, with visualizations provided in Fig. 24.

For the comparison method NFGP Yang et al. (2021), we manually identified handle points for five time steps, defined the rotation and translation between these points, and trained the model for each pair sequentially, requiring five separate training iterations.

## A.13 CHANGE GENUS EXAMPLE AND DIRECTLY DEFORMING TRIANGLE MESHES

In this section, we demonstrate that our method effectively handles genus-changing deformations. Additionally, we show that our approach can directly deform triangle meshes by treating mesh

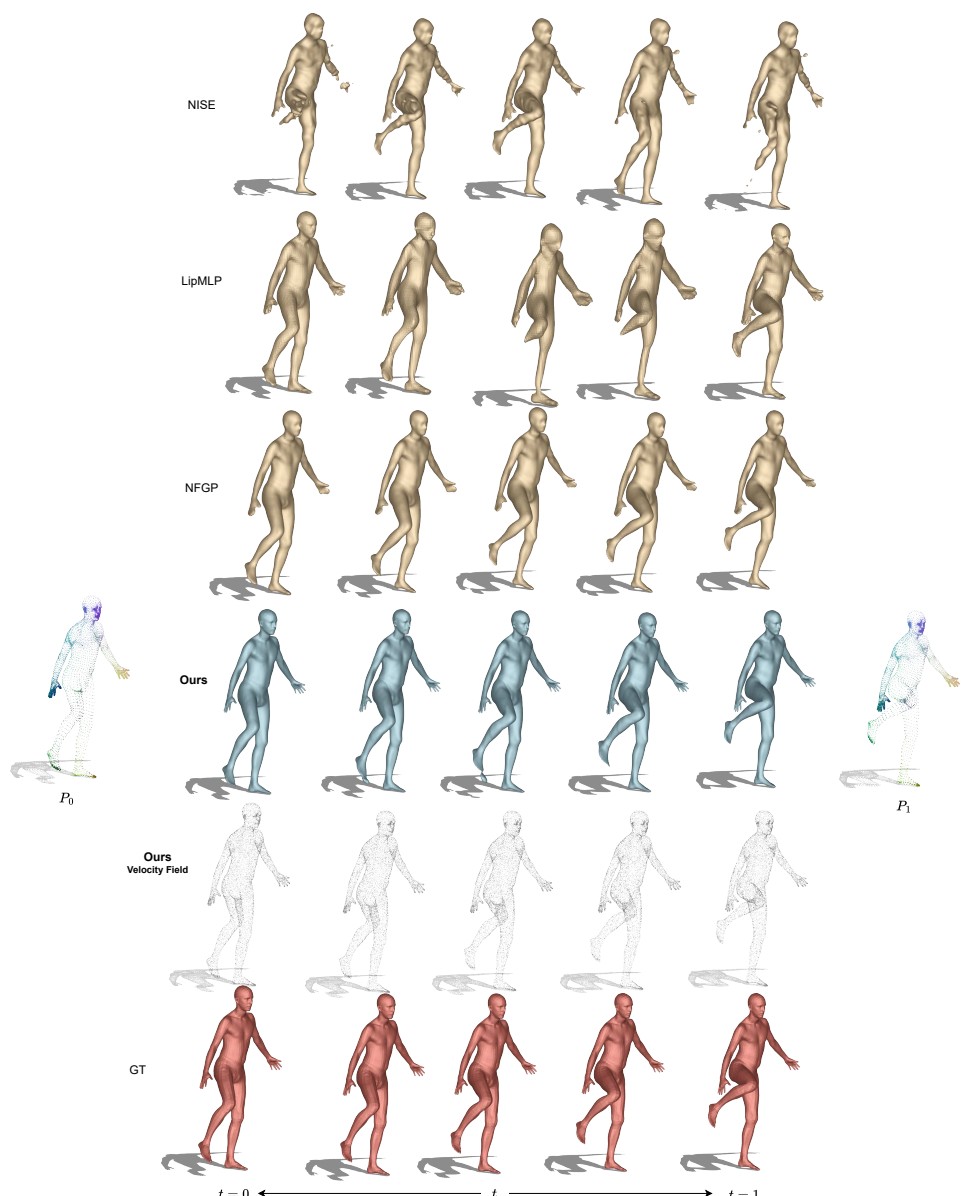

Figure 24: **Visualization results of 4D-Dress** Wang et al. (2024). The dataset consists of high-frequency meshes with deformations. We sample points from the first and fifth meshes as the source and target inputs, respectively, and recover the intermediate shapes. Correspondences are visualized using colored points ($P_0$ and $P_1$), with the bottom row showing the ground truth SMPL Loper et al. (2015) model. Additionally, we visualize our velocity field as a sequence of point cloud deformations in the second row from the bottom.

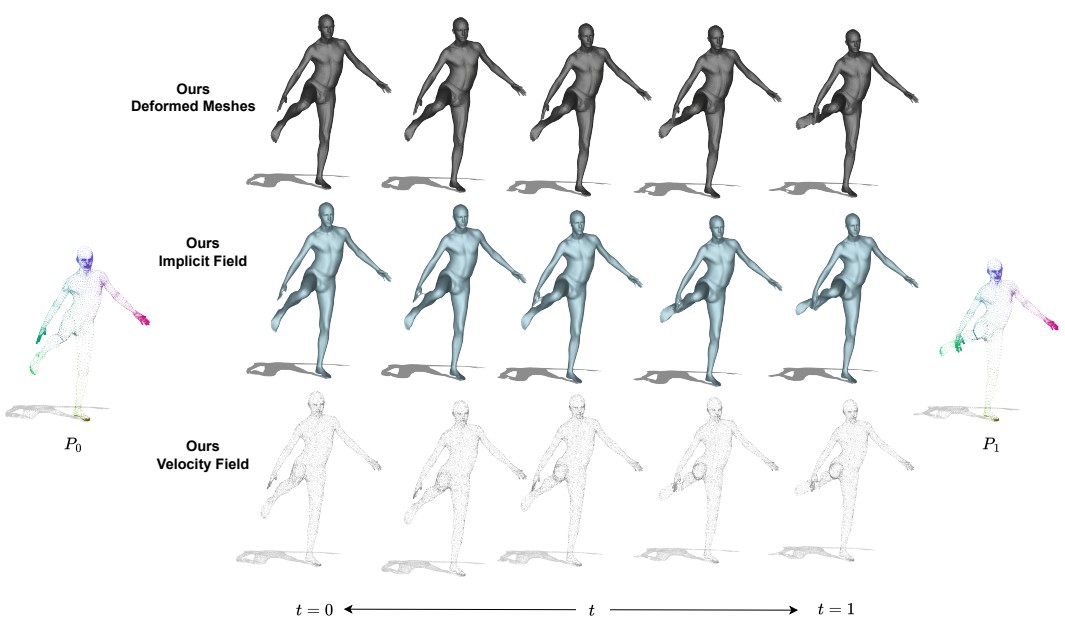

Figure 25: **Change genus case.** In this example, we present a case of genus change, showcasing how the implicit representation enables arbitrary topological transformations. Furthermore, we demonstrate that our method can directly deform triangle meshes using a velocity network by treating mesh vertices as point cloud points. Notably, there is a clear resolution difference between the deformed meshes and the reconstructed meshes produced by our implicit network.

vertices as points in a point cloud. Notably, the original triangle meshes contain $6,890$ vertices. Leveraging our implicit network, which supports arbitrary resolution for output meshes, we rendered the deformed meshes with approximately $106,000$ vertices.

## A.14 No Correspondences and Partial Correspondences

In this section, we highlight scenarios where no correspondences or only partial correspondences are available. The results in Fig. 26 demonstrate that our velocity field remains consistent, and the implicit network does not degenerate, unlike NISE Novello et al. (2023) or LipMLP Liu et al. (2022). Remarkably, even without correspondences, our method can still recover reasonable deformations between the two input point clouds, as shown in Fig. 27. We attribute this robustness to the joint training of our implicit network with strong physical constraints. However, we observe that, in the absence of correspondences, the deformations are less smooth compared to cases with correspondences, and some artifacts tend to appear around the recovered surfaces.

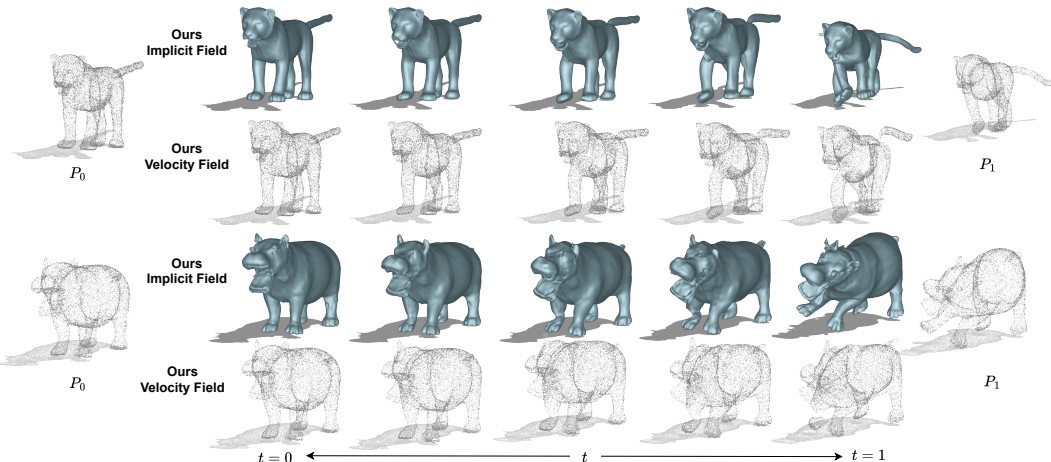

Figure 26: **No correspondences case.** In this example, we present a case with no correspondences. Remarkably, our method can still handle certain deformations, thanks to the volume-preserving constraint, smoothness constraint, and joint training with the implicit network.

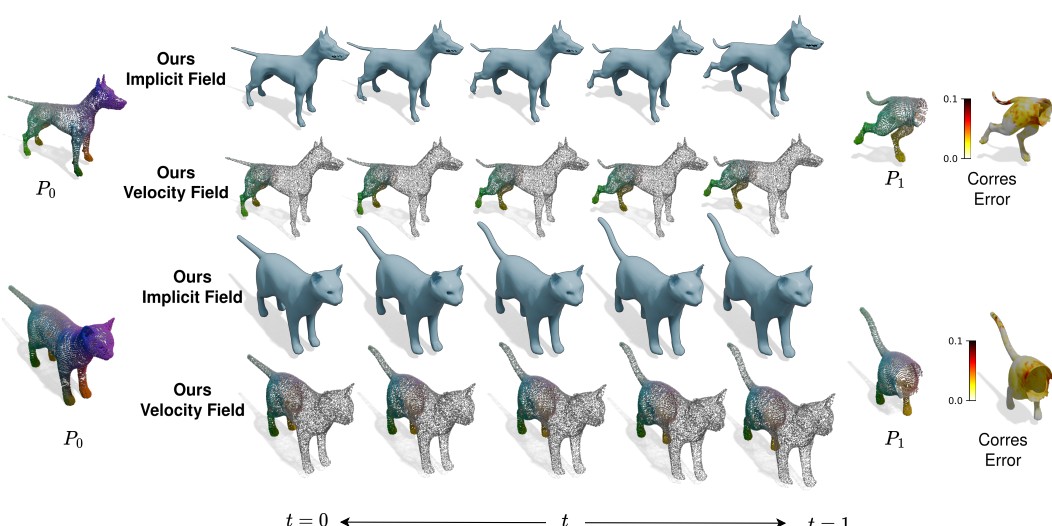

Figure 27: **Partial shape interpolation.** We demonstrate partial correspondences using a partial target point cloud. The correspondence error is measured as the misaligned geodesic distance relative to the ground truth, with the total mesh area normalized to 1. Despite the incomplete target shape, our method successfully recovers the intermediate meshes.

