# OpenReview forum: "Implicit Neural Surface Deformation with Explicit Velocity Fields"
_ICLR.cc/2025/Conference — ICLR 2025 Poster_

### Official Review · Reviewer_NAfz · 2024-10-30

**Soundness:** 3
**Presentation:** 3
**Contribution:** 3
**Rating:** 8
**Confidence:** 3

**Summary:**

Leveraging the Lagrangian approach in fluid mechanics, the authors introduce a novel method for implicit surface deformation. It models point trajectories during surface deformation using explicit velocity fields, enabling the recovery of physically plausible intermediate shapes through a modified level-set function. Unlike previous implicit methods, this approach requires only a sparse set of correspondences between the source and target point clouds in the unsupervised loss computation, eliminating the need for inefficient supervision from intermediate shapes. Experiments cover both intra- and inter-category shape deformations, as well as point clouds of different quality (sparse/incomplete).

**Strengths:**

- The paper is well-written, clear, and easy to follow.

- The authors provide detailed derivations for formulating the Modified Level-Set Equation from the ordinary differential equation of the velocity field.

- The proposed method deforms source and target surfaces using a loss function that effectively avoids dependence on intermediate shapes, and requires only a sparse set of correspondences between the source and target point clouds.

- The method demonstrates efficiency in handling pose deformation within the same category and non-rigid deformation across different categories, and is flexible to incomplete and sparse point cloud inputs.

- Ablation studies are provided to validate robustness of the method to different correspondence qualities.

- Multiple visualizations are provided for qualitative evaluations of the method.

**Weaknesses:**

- Correspondences between the source and target shape have to be provided, which are not always available. Establishing correspondences between cross-category shapes might be challenging.

- It is more informative if the correspondences utilized in deforming each surface pair can be visualized as well in some figures.

- The readability of Figure 2 and Figure 7 should be improved.

**Questions:**

- How are the sparse correspondences selected for each deformation? Do they have to cover the entire shape, or is the method still effective if the correspondences are nested in certain regions of the surface?

---

> ### Author Response · Authors · 2024-11-19
>
> Thank you very much for the positive feedback!!! We sincerely thank the reviewer for the thoughtful and encouraging feedback on our work. We are delighted to hear that the reviewer appreciated our methodology and the significance of our contributions. Regarding the weaknesses and questions raised by the reviewer, we address them as follows:
>
> **Correspondences:** It is indeed the primary limitation of our work. However, as demonstrated in section 4.3, Fig.11, our pipeline can integrate with a mesh-matching approach to obtain the required correspondences. Compared to using mesh shape matching combined with interpolation, such as the state-of-the-art method in [1], our approach (as shown in Appendix Section A10) offers the ability to interpolate incomplete shapes and handle topologically inconsistent shapes.
> Additionally, we have updated the appendix with experiments where correspondences are obtained from [2] (as [1] cannot handle partial shapes). These updates include error maps of the correspondences, showcasing how our method can integrate with existing approaches to address the correspondence challenge effectively.
>
> [1] SMS: Spectral Meets Spatial: Harmonising 3D Shape Matching and Interpolation (CVPR 2024). Cao, Dongliang and Eisenberger, Marvin and El Amrani, Nafie and Cremers, Daniel and Bernard, Florian
>
> [2] Unsupervised learning of robust spectral shape matching (SIGGRAPH 2023) Cao, Dongliang and Roetzer, Paul and Bernard, Florian
>
> **Readability of Fig. 2 and Fig. 7:** We thank the reviewer for the suggestion, we will include a larger version in the Appendix. Additionally, we will re-render Fig. 7 to improve its clarity.
>
> **Correspondence distribution/selection:** The correspondences in our method are selected uniformly, which means they more or less cover the entire shape under the current settings. To further explore different scenarios, we added two new examples:
>
> 1) ***Partial Shape Interpolation:*** In this case, only a partial shape of the target and partial correspondences are available. In the updated manuscript Fig. 27, we present examples of partial shapes along with the correspondence error. The correspondences were obtained using [2], and our method successfully handles these cases.
>
> 2) ***No Correspondences:*** We also examined a scenario where no correspondences are provided. Even in the absence of correspondences, our method recovers reasonable deformations. These results are shown in the updated manuscript Fig. 26.
>
> That said, we acknowledge the limitations of these cases. For partial shapes, correspondences are available in regions where the shape deforms, while for the no-correspondence case, we only tested small deformations. Based on our observations, our method remains robust under relatively small deformations when correspondences are nested. Moreover, it appears that correspondences are only necessary in regions where deformation occurs.

---

> > ### Comment · Reviewer_NAfz · 2024-11-25
> >
> > I would like to thank the authors for their reply. First, I am satisfied with the contribution of this work, as it indeed eliminates the need for supervision from intermediate shapes during deformation, despite the requirement for correspondences between source and target point clouds.
> >
> > Second, I speculate that the interesting 'line segment effect' raised by Reviewer qpsz and the 'larger deformation' raised by Reviewer RFX6 are related, as the velocity field is probably fitting paths where straight lines are preferred during the interpolation, resulting in the method's inability to handle larger deformations.
> >
> > Nonetheless, I vote for acceptance of the work, and am interested in seeing how the discussion on large deformations develops.

---

> ### Author Response · Authors · 2024-11-25
> **Thank you for your positive feedback**
>
> We sincerely appreciate the reviewer's positive feedback.
>
> - Regarding the **"Line segment effect"**, we have added a new figure (Fig.30) in our supplementary material (Please download **supplementary material zip file** and check **4738_SuppMat.pdf**).  We visualized the trajectory of a linear interpolated movement (first row) and the point trajectory from our velocity field (second row). It is evident that our velocity field generates a curved movement that preserves the length of the leg, whereas linear interpolation shortens the leg length to achieve linear motion.
>
> - Regarding the **Large deformation**: from the results of Fig.30, we do not think that our method produces a velocity net that prefers linear movements. Regarding the very large deformation, we believe that under our current setting, the problem is hard to fully constrain.  We would like to point out that our method is different from a **template or skeleton-based** method that can deal with very large and different movements, such as [1, 2]. These methods focus on the articulation of the human movement and have much stronger assumptions than our method. We proposed a more general framework to deform the implicit field which makes the deformation physically plausible. However, as shown in Fig. 22, our approach successfully handles relatively large deformations, even in cases with topological inconsistencies.
>
> [1] Human Motion Diffusion as a Generative Prior (ICLR 2024) Yonatan Shafir and Guy Tevet and Roy Kapon and Amit H. Bermano
>
> [2] Learning dynamic relationships for 3d human motion prediction (CVPR 2020) Cui, Qiongjie, Huaijiang Sun, and Fei Yang.
>
> We sincerely thank the reviewer once again for their insightful feedback. We believe our work has the potential to inspire further research in this direction. Your positive review is both encouraging and greatly appreciated.

---

### Official Review · Reviewer_RFX6 · 2024-11-01

**Soundness:** 3
**Presentation:** 3
**Contribution:** 3
**Rating:** 6
**Confidence:** 3

**Summary:**

This paper focuses on the task of learning continuous intermediate surface deformations between two given 3D point clouds. An end-to-end framework built upon fluid mechanics and physical constraints is designed to learn the time-varying implicit field via an explicit velocity field. Various comparison experiments and ablation studies are conducted to show the effectiveness and characteristics of the proposed learning approach.

**Strengths:**

- The theoretical formulations are well-motivated and technically sound, and the learned deformation results look smooth and reasonable.
- The method is able to handle rigid and non-rigid deformations without intermediate shape supervision.

**Weaknesses:**

- One critical issue is the setting that a certain ratio of ground-truth point-to-point correspondences must be provided (although might be partially noisy). In fact, this is a very strong prior, especially when the points that are given GT correspondences are scattered around the whole shape surface. For example, during training 20,000 points are sampled from the surface, 5%-20% of them (1000-4000 points) are corresponded. This prior information makes the task itself much easier. From this perspective, I do not agree with the claim that the proposed learning framework is fully “unsupervised”.

- The experimental results suggest that the input shapes have similar geometries or poses and the method cannot handle very large deformations.

**Questions:**

As for the divergence-free velocity field, is it reasonable to constrain that the total mass inside surface always stay the same, especially when we deform two objects of distinct types and/or categories?

Figure 6 presents an interesting example of handling incomplete inputs. What would happen if the input model becomes "broken," for example, if one paw is separated from the dog's leg? Is your method robust enough to handle such scenarios?

I suggest adding visulizations of the learned velocity fields.

Since the method is SDF-driven, it cannot handle surfaces with large boundaries, such as the clothes models in the DeepFashion3D dataset. Nevertheless, it would be beneficial to discuss this issue.

Can your method handle inputs of different topologies? For example, a standing man with genus 0 and another person with hands on the waist, which is of genus 2.

The authors claim that the intermediate shapes are physically plausible; however, all evaluation metrics provided are geometric. Additionally, the velocity loss, which includes a smoothness term and a divergence-free term, is not directly related to physical principles. Therefore, this claim requires better justification using quantitative physical measures.

---

> ### Author Response · Authors · 2024-11-19
>
> We would like to express our sincere gratitude to the reviewer for the insightful feedback on our manuscript. We greatly appreciate the positive remark about our method and the further experiment suggestions, which will enhance the robustness and comprehensiveness of our study. We included the additional results and updated our manuscript. In response to the weaknesses and questions raised by the reviewers, we address them as follows:
>
> **Unsupervised:** We agree with the reviewer and we apologize for the misleading word “unsupervised”, which means only no supervision on intermediate shapes. We updated the PDF to clean up the misunderstanding.
>
> **Correspondences:** It is indeed the primary limitation of our work. However, as demonstrated in section 4.3, Fig.11, our pipeline can integrate with a mesh-matching approach to obtain the required correspondences. Compared to using mesh shape matching combined with interpolation, such as the state-of-the-art method in [1], our approach (as shown in appendix Section A10) offers the ability to interpolate incomplete shapes and handle topologically inconsistent shapes.
> Additionally, we have updated the appendix with experiments where correspondences are obtained from [2] (as [1] cannot handle partial shapes). These updates include error maps of the correspondences, showcasing how our method can integrate with existing approaches to address the correspondence challenge effectively.
>
> [1] SMS: Spectral Meets Spatial: Harmonising 3D Shape Matching and Interpolation (CVPR 2024). Cao, Dongliang and Eisenberger, Marvin and El Amrani, Nafie and Cremers, Daniel and Bernard, Florian
>
> [2] Unsupervised learning of robust spectral shape matching (SIGGRAPH 2023) Cao, Dongliang and Roetzer, Paul and Bernard, Florian
>
> **Larger deformation:** It is true that our method occasionally struggles with very large deformations. However, as shown in Fig. 22, our approach successfully handles relatively large deformations, even in cases with topological inconsistencies. We would like to point out that our method is different from a template or skeleton-based method that can deal with very large and different movements, such as [1, 2]. These methods focus on the articulation of the human movement. We proposed a more general framework to deform the implicit field which makes the deformation physically plausible. Furthermore, we have included additional experiments in the appendix to demonstrate the capability of our method in addressing extreme partial shape deformation scenarios.
>
> Despite its current limitations, we believe our method represents a strong starting point for direct implicit surface deformation using point cloud inputs and hope it will inspire more follow-up works.
>
> [1] Human Motion Diffusion as a Generative Prior (ICLR 2024) Yonatan Shafir and Guy Tevet and Roy Kapon and Amit H. Bermano
>
> [2] Learning dynamic relationships for 3d human motion prediction (CVPR 2020) Cui, Qiongjie, Huaijiang Sun, and Fei Yang.
>
> **Volume Preserving:**  This is an insightful observation by the reviewer. Indeed, we set the weight for the divergence-free loss to zero when performing cross-category interpolation, as mentioned in Lines 352-353 of the manuscript. We emphasize that we enforce volume preserving as a loss, not by constructing the velocity field with a divergence-free structure, such as [3], which makes it less flexible to handle non-volume preserving cases.
>
> [3] Divergence-Free Shape Interpolation and Correspondence (CGF 2019). Marvin Eisenberger and Zorah Lähner and Daniel Cremers
>
> **Broken input:** Our implicit neural network is initialized as a sphere, which naturally tends to close gaps or holes on surfaces. As a result, small broken features, such as incomplete legs, are likely to be reconstructed as whole, continuous structures. However, if the broken parts are too far away from each other, our method might struggle with it. We will include this in our paper’s discussion section. Instead of this situation, we demonstrate a different form of shape completion: our method’s ability to handle cases where the target shape is only partial. For details, please refer to the updated manuscript Fig. 27 in the appendix.
>
> **Visualization velocity field:** We thank the reviewer for the thoughtful suggestion. In response, we have added visualizations of the new experiments, including the velocity fields and the deformed point clouds. Please refer to the updated manuscript Appendix Fig. 24-27 for details.
>
> **Large Boundaries:** We thank the reviewer for pointing this out. SDF indeed has its limitations, and we will include this discussion in the limitations section of our work.

---

> > ### Author Response · Authors · 2024-11-19
> >
> > **Different genus:** Yes, our method can handle inputs of different topologies. In fact, this is a key motivation for using implicit representations. In Fig. 22 of the Appendix, we demonstrate examples of topological inconsistencies. Additionally, in Fig. 25, we present another example where a standing man raises his leg and holds it, showcasing the flexibility of our approach.
> >
> > **Quantitative physical measures:** The physical plausibility involves volume-preserving and spatial smoothness. Indeed the physical plausibility lacks a clear mathematical definition, so it is hard to quantify. In mesh-based methods, usually, people show that the geodesic distance of two corresponding points is preserved to show physical plausibility. However, in our method, we do not have fixed vertex-to-vertex correspondences across generated meshes as our meshes are represented by SDF. Thus, we mainly show the physical plausibility by visual results.  Another way is to compare the generated mesh to ground truth meshes.
> >
> > To better evaluate our method, we add an experiment using the 4D-Dress dataset [1], which contains high-frequency meshes capturing human movements. For our evaluation, we selected a mesh pair with four intermediate meshes, using the first and last as the source and target inputs. The four intermediate meshes serve as ground truth. We then computed the Chamfer Distance (CD) and Hausdorff Distance (HD) to quantitatively evaluate our method.
> >
> > To further demonstrate that our method is volume-preserving, we calculated the standard deviation of the surface area across all interpolated meshes. This ensures that the meshes are neither overstretched nor shrunk during deformation. The results are presented in Table 1, with visualizations shown in Fig. 24.
> >
> > [1] 4d-dress: A 4d dataset of real-world human clothing with semantic annotations (CVPR 2024) Wenbo Wang and Hsuan-I Ho and Chen Guo and Boxiang Rong and Artur Grigorev and Jie Song and Juan Jose Zarate and Otmar Hilliges.

---

> > > ### Comment · Reviewer_RFX6 · 2024-11-25
> > >
> > > Dear authors, Thank you for your detailed responses, which have addressed my concerns. I have no further questions and believe this is good work.

---

> > > > ### Author Response · Authors · 2024-11-26
> > > > **Thank you for your feedback**
> > > >
> > > > Thank you very much for your time to review our paper and the positive feedback!
> > > > We have included more experiments in the updated manuscript and supplementary material (please download the **supplementary material zip** file to see the newest version):
> > > >
> > > > 1) **Quantitative evaluation for physically plausible intermediate shapes**. See Fig. 24 and Tab. 1. Together with the visualization of our velocity field.
> > > > 2) **Visual example with topological changes** (from genus 0 to genus 1) and a **direct deformation of triangle meshes**, see Fig 25.
> > > > 3) The experiment of **no correspondences** case, see Fig 26.
> > > > 4) **Partial correspondences and partial target shape with correspondence error map**. See Fig. 27.
> > > > 5) A **Video** (dress4d_video.mp4) in Supplementary Material zip file to demonstrate the interpolated mesh in Fig.24. (Please download our new **Supplementary Material**).
> > > > 6) A **Video** (faust_video.mp4) in Supplementary Material zip file to demonstrate the interpolated mesh in Fig.3. (Please download our new **Supplementary Material**).
> > > > 7)  **Quantitative and qualitative evaluation on noisy correspondences**: See Fig.28 and Tab.2 in our new **Supplementary Material**.
> > > >
> > > >
> > > > We hope these additions provide more detailed results, demonstrate the capabilities of our method, and further address the reviewers’ concerns. If our response has adequately addressed your concerns, It would be appreciated if you could consider raising the score for our paper. We thank you again for your effort in reviewing our paper.

---

### Official Review · Reviewer_qpsz · 2024-11-02

**Soundness:** 3
**Presentation:** 3
**Contribution:** 3
**Rating:** 6
**Confidence:** 5

**Summary:**

This paper proposed a self supervised method for frame interpolation between two given point clouds. Technically, the proposed method models the point movement using an explicit velocity field and transform the time-varying implicit field through modified level-set equation. Experiments on various datasets show its effectiveness.

**Strengths:**

* The writing is easy to read and follow.
* The problem modeling is clear and the formula derivations are also correct.
* The proposed method achieves better performance than baseline methods at different datasets.

**Weaknesses:**

* As shown in Fig. 4, the trajectory of each point is assumed as a line segment, making the shape different than the tow given shapes.
* The datasets used in the experiment do not consists of complex diformation. Can the proposed method deal with these conditions?
* There is no efficiency analysis about the proposed methods and baseline methods, which is important in practice.
* There is no experiment about reconstruction accuracy under different network sizes.

**Questions:**

* Can the proposed method be used to process non-watertight surfaces, such as clothes, by replacing the SDF with UDF?
* Should Eq. 4 consist of $\phi(x,t)=x^1$, where $x^1$ is the endpoint of the trajectory.
* Why not include Eq. 3 in the loss function?
* Given two shapes, there are many methods to calculate the flow of each vertices/points. Once the flow calculated, the frames between them are easy to interpolate. What is the advantages over these methods?
* Can the proposed method be used for triangle mesh directly?
* If replace the marching cubes with dual countouring, would the reconstruction increase?

---

> ### Author Response · Authors · 2024-11-19
>
> We would like to thank the reviewers for the thorough and insightful feedback on our manuscript. We are especially appreciative of the positive comments. Additionally, we are grateful for the suggestions. Regarding the weaknesses and questions raised by the reviewer, we address them as follows:
>
> **Line segment in Fig. 4:** We apologize for any confusion. To clarify, our method does not rely on any pre-defined (or linear) path when deforming shapes. Instead, we work directly with a source point cloud and a target point cloud. As shown in Fig. 4, our meshes at  $t = 0$  and  $t = 1$  align well with the respective point clouds, illustrating the effectiveness of our approach.  In case we misunderstood the reviewer’s concern, we kindly ask to clarify what is meant by “line segment”, so that we can address the concern more thoroughly.
>
> **No complex deformation:** We believe our experiments encompass a diverse range of scenarios, including non-isometric deformations and incomplete shape deformations. Non-isometric deformations in our context involve not only gesture changes but also category changes.  If the reviewer is referring to examples with larger movements, we kindly direct their attention to Fig. 22 in the appendix. Additionally, we have included more examples of larger deformations to further demonstrate the robustness and versatility of our method in newly added experiments Fig. 25-27.
>
> **Efficiency Analysis:** We apologize for not emphasizing the efficiency aspect more clearly. However, we emphasize that mention the running time in Section 4.1 (Line 339-342 in the original manuscript and Line 342-347 in the updated version). As stated in the paper, NISE requires approximately 1.5 hours per deformation step, excluding the pre-training time for the SDF networks of the source and target meshes. while NFGP takes around 15 hours per step (over 75 hours for five steps). In contrast, our method is significantly more efficient, requiring only 1/5 of the time compared to NISE, which takes an additional 20 minutes. To provide further clarity, we have added a section in the appendix detailing the training processes for both the comparison methods and our approach.
>
> **Ablation of Network Size:**  We have not done any network size ablation because our main contribution is not focusing on network architectures and we just use two simple MLP with 8 layers, as mentioned in section 4. Line 286-287. However, we will add network size ablation with a small node size and fewer layers. We will update the results in the following days.
>
> **Use UDF instead of SDF:** The short answer is: yes, it can be used for non-watertight surfaces. The level set equation operates on level-sets, meaning it can be generalized to UDF. However, from a mathematical perspective, SDF captures more intrinsic properties of the surface. If the goal is to represent an open surface, we sincerely recommend considering an uncertainty-aware SDF approach [1]. In this method, the open surface is treated as a pseudo-closed surface, where the open regions are assigned a weight of 0. During the marching cubes process, surfaces are generated only for regions with weights greater than 0. Alternatively, a hybrid representation [3] could also be considered.
> From a practical standpoint and based on our limited experience with UDF, it often struggles with identifying zero-crossings. While some works [2] propose methods to address this issue, they require learning to fit the UDF locally to an SDF, which can add complexity.
>
> [1] Enhancing Surface Neural Implicits with Curvature-Guided Sampling and Uncertainty-Augmented Representations, (ECCVW 2024) L. Sang and A. Saroha and M. Gao and D. Cremers
>
> [2] Neural Surface Detection for Unsigned Distance Fields (ECCV 2024) Federico Stella and Nicolas Talabot and Hieu Le and Pascal Fua
>
> [3] HSDF: Hybrid Sign and Distance Field for Modeling Surfaces with Arbitrary Topologies (Neurips 2022) Wang, Li and Yang, Jie and Chen, Weikai and Meng, Xiaoxu and Yang, Bo and Li, Jintao and Gao, Lin
>
> **Equation.4:** Yes, the reviewer is correct. In our setup, we not only fix the starting point but also fix the ending point. However, since we were introducing the velocity ODE equation and aimed to maintain the form of the Cauchy initial problem for the ODE [1], we specified only the initial condition in Eq. (4). Based on the reviewer’s feedback, we have revised the corresponding text and added the condition \phi(x,1) = x^1 to clarify this point.
>
> [1] The Cauchy Problem for Ordinary Differential Equations. In: Parametric Continuation and Optimal Parametrization in Applied Mathematics and Mechanics. (Springer 2003). Shalashilin, V.I., Kuznetsov, E.B.

---

> > ### Author Response · Authors · 2024-11-19
> >
> > **No Eikonal Loss:** This is because we adopt MLSE, which inherently incorporates the Eikonal loss into the level-set equation. This approach allows us to enforce a continuous, integrated constraint on the level-set, eliminating the need for a reinitialization process, as explained in Section 3.3 (Lines 226-245). Additionally, we observed that using the original level-set equation with a separate Eikonal loss occasionally led to mesh degeneration during recovery, as discussed in Sections 4.3 and A.6 of the appendix.
> >
> > **Advantages of our method:** To the best of our knowledge, most methods for computing flows, such as [1] and [2], are mesh-based. These approaches inherit the common limitations of explicit shape representations, including difficulties in handling topological changes and the dependence on specific parameterizations. As discussed in Lines 94-101 of the manuscript (explicitly elaborated), these drawbacks limit their applicability. Furthermore, a detailed comparison is provided in the original manuscript Appendix Section A.10.
> > In contrast, our method is capable of handling topological changes. Additionally, our method effectively addresses interpolation for incomplete shapes compared to current SOTA method [1], highlighting a unique strength of our approach. We can handle more challenging cases such as partial inputs (Fig. 6, Fig. 27), Genus changing cases Fig. 25.
> >
> > [1] SMS: Spectral Meets Spatial: Harmonising 3D Shape Matching and Interpolation (CVPR 2024). Cao, Dongliang and Eisenberger, Marvin and El Amrani, Nafie and Cremers, Daniel and Bernard, Florian
> >
> > [2] NeuroMorph: Unsupervised Shape Interpolation and Correspondence in One Go (CVPR 2021) M. Eisenberger and D. Novotny and G. Kerchenbaum and P. Labatut and N. Neverova and D. Cremers and A. Vedaldi
> >
> > **Move triangle meshes:** Yes, this is possible. The mesh vertices can be treated as points in the point cloud, allowing us to directly deform the point cloud using the velocity network while preserving the original mesh structure (faces and edges). To illustrate this, we have added an example of direct mesh deformation in the appendix (Fig. 25).
> >
> > **Other meshing method:** There exist more sophisticated methods such as dual contouring.  In principle, any meshing method capable of extracting meshes from an SDF could be integrated with our approach. However, mesh extraction from SDFs is not the focus of our work nor a primary contribution of this study.

---

> > ### Comment · Reviewer_qpsz · 2024-11-22
> > **Some question about trajectory.**
> >
> > Thank you for your reply! Most of my comments have been addressed.
> >
> > Now, let’s discuss the trajectories in more detail.
> > The primary application of this paper is frame interpolation between two given frames, which makes the trajectories of individual points on the shapes crucial. For example, consider Fig. 3, where the leg is rotating around a joint. In this case, the trajectory of the points on the foot should follow a circular path. However, the results in Fig. 3 indicate that the proposed method assumes the trajectory to be a straight line segment, leading to distortions in the interpolated frames.

---

> ### Author Response · Authors · 2024-11-22
> **Trajectory Clarification**
>
> We now understand what the reviewer means by “line segment.” The reviewer is correct that we estimate the trajectory of each point on the shape through our velocity field. However, we do not impose any prior assumption, such as requiring points to move along straight lines. Instead, **we constrain the movement using spatial smoothness and volume-preserving terms to ensure the motion remains physically plausible.**
>
> We believe the “line segment effect” observed may be due to the limitations of static visualizations. To address this, we have included two **videos** (associated with Fig. 24, and Fig.3) to illustrate the movement better and highlight our method's results. Please download the **Supplementary Material zip file**. Additionally, Fig. 27 was produced using the same loss function as Fig. 3, and it demonstrates that our method does not produce linear movement but instead achieves physically plausible trajectories (see the stretching of the legs and tails of Fig.27)
>
> We hope the added video clarifies the behavior of our method.
>
> Update: We also uploaded the network size ablation in the **supplementary material zip file**, please download the zip file to see the new added experiment results.

---

> > ### Comment · Reviewer_qpsz · 2024-11-25
> >
> > Thanks for your reply.
> > * In Fig. 3, the distortion of the foot in the reconstructed interpolated surface is not caused by the static visualizations. If the reconstruction were accurate, each frame would not exhibit such distortion.
> > * For instance, in Fig. 27, the dog on the right side of the frame at t=0 shows distortion in its foot, which I believe is caused by the "line segment effect." I think it is impossible to accurately predict a circular path using only two frames, which is the main weakness of the proposed method. Could this issue be resolved if more frames were provided?

---

> > > ### Author Response · Authors · 2024-11-25
> > > **Linear Segment Clarification**
> > >
> > > We sincerely thank the reviewer for their further clarification.
> > >
> > > **Regarding the "Line Segment Effect"**
> > > We see how the pointed images may lead to the intuition that our method leads to a trajectory close to linearity. However, we do believe our energies promote more complex behaviors than linear ones since a pure linear interpolation would be not volume-preserving. To show this, we report a further visualization in Fig. 30 (Please download our **new supplementary material zip** file and check the **4738_SuppMat.pdf** file). We visualized the trajectory of a linear interpolated movement (first row) and the point trajectory from our velocity field (second row). It is evident that our velocity field generates a curved movement that preserves the length of the leg, whereas linear interpolation shortens the leg length to achieve linear motion.
> > >
> > > However, we agree with the reviewer that our method interpolates the movement along the shortest distance path (i.e., we cannot “generate” or “predict” additional intermediate movements) and that the velocity field simulates movement linearly over an infinitesimal distance. As described in Section 3.2 of our paper, the integration of the velocity field (Eq. 8) is implemented using Euler steps, which assume that within a small time increment $\delta t$, the movement is linear. This approach effectively breaks down rotational or non-linear distortion into infinitesimally small linear segments.
> > >
> > > **Regarding Circular Paths**: We acknowledge the reviewer's point regarding scenarios with circular paths. If the starting and ending positions are the same, our method cannot predict such circular paths in the middle. This limitation arises because our approach does not incorporate any priors, or pre-trained data (as in generative or diffusion models).
> > >
> > > However, with the inclusion of additional frames, such as keyframes within the circular motion, our method demonstrates the ability to interpolate movements more accurately.
> > >
> > > Despite this limitation, we have shown that our method:
> > >
> > > 1). Introduced a novel directly deforming implicit field via the modified level-set equation framework.
> > >
> > > 2).  Produces significantly better results compared to baseline methods,
> > >
> > > 3).   Handles both isometric and non-isometric deformations effectively, and
> > >
> > > 4).  Efficiently manages partial shape interpolation.
> > >
> > > We believe these achievements underscore the significance of our contributions.
> > >
> > > We thank the reviewer once again for your thorough discussion and insightful feedback. We hope this answer addresses the reviewer's concerns.

---

> > > > ### Comment · Reviewer_qpsz · 2024-11-26
> > > >
> > > > Thanks for your reply, and I think the 'Line Segment Effect' could be your future work.
> > > > On the other hand, I wonder how about the ablation study about the network size? Is it updated on the revised manuscript?

---

> > > > > ### Author Response · Authors · 2024-11-26
> > > > > **Network size ablation**
> > > > >
> > > > > Thank you very much and we will certainly continue to improve our work.
> > > > >
> > > > > We have included the network ablation in the **Supplementary Material**, we kindly ask the reviewer to download the **supplementary material zip file** and check the **4738_SuppMat.pdf** files, ablation is included in Tab.S 3 and Fig. S 29.
> > > > >
> > > > > We would like to thank the reviewer for the valuable comments and suggestions. As the deadline for revising our PDF file and adding visual results is approaching, we want to check if there are any additional experiments you would like us to include. If so, please let us know, and we will do our best to incorporate them before the deadline.
> > > > >
> > > > > If our response has adequately addressed your concerns, It would be appreciated if you could consider raising the score for our paper. We thank you again for your effort in reviewing our paper.

---

> > > > > > ### Comment · Reviewer_qpsz · 2024-11-29
> > > > > >
> > > > > > Thanks for your reply, and it is wonderful to have a discussion with you. Obviously, other reviewers and I admit your contribution to the field of dynamic object modeling. I will raise my rating and hope the paper accepted.

---

> > > > > > > ### Author Response · Authors · 2024-11-29
> > > > > > > **Thank you very much**
> > > > > > >
> > > > > > > Thank you very much for your feedback and all your interaction with us. We are deeply grateful for the time and effort the reviewer has invested in evaluating our work.

---

### Official Review · Reviewer_hBUT · 2024-11-04

**Soundness:** 3
**Presentation:** 3
**Contribution:** 3
**Rating:** 6
**Confidence:** 4

**Summary:**

The paper proposes a method to simultaneously recover the implicit neural representation of two given point cloud inputs, together with time-varying intermediate shapes between them. The point movement is modeled using an explicit velocity field regularized with smoothness and divergence-free constraints, and the shape is implicitly deformed using the modified level-set equation. The method is evaluated on four datasets, demonstrating its ability to model both pose and non-rigid deformation, as well as its robustness to incomplete and sparse input.

**Strengths:**

- **The modified level-set equation**: The paper modifies the Level-set equation by introducing an additional term (Equ.12). This modification ensures both the velocity field and implicit function are supervised to enforce the Eikonal constraint at each step. This approach is novel and effective in this context.
- **Performance**: The method demonstrates a better capability to recover physically plausible intermediate shapes compared to the given baselines.
- **Robustness to Incomplete and Sparse Input**: The proposed method is robust to the size of the target input point cloud. Unlike other methods, it can accurately recover the target shape and generate plausible intermediate shapes, even when the input is ten times sparser.

**Weaknesses:**

- **Motivation**: Please clarify the motivation for this setting. Why is it important to recover physically plausible intermediate shapes? Why is it reasonable to assume that correspondances are available?
- **Correspondances**: While an ablation of the sparsity of the correspondences is available in the supplementary materiel, it’s not clear what happens when correspondances are not available i.e without the last term of Equ.15. On the other hand, can other methods rely on the correspondance information? For example, could this constraint be embedded in NISE and how does it perform in this case ?
On the other hand, one numerical result that can strenghten the paper is the average error of the method using correspondences from Cao et al. (2024a) . This should be added to table (a). In addition, it would be helpfull to also show the result with GT correspondences  in Fig.11 to better see the difference.
- **Ablation of the modified level-set equation ablation**: Figure 8 is a bit surprising. Why is the OLSE baseline not able to recover the final shape while this constraint is explicitly enforced in the loss function ? Is the volume preserving term used in this experiment ?

**Questions:**

- What is the main motivation for this setting ?
- How  robust is the method to the failures of point-registration techniques, when ground-truth correspondences are not available ?

---

> ### Author Response · Authors · 2024-11-19
>
> We thank the reviewer for their insightful questions and positive feedback. We particularly thank the reviewer for suggesting better motivation reseasoning, an experiment with no correspondences, and a better explanation about OLSE vs. MLSE. Regarding the weaknesses and questions raised by the reviewer, we address them as follows:
> **Motivation**:
>
> ***Why are physically plausible Intermediate shapes necessary:***
>
> We answer this question in two aspects:
> 1) why do we need to interpolate intermediate shapes?  The real world is continuous. However, in most cases, we only have discrete observations. To recover the continuous movements and deformation that happen in the real world, it is useful and important to be able to interpolate between discrete observations. For example, some human movements are recorded by 1 FPS (frames per second) sequence and we would like to have at least 30 FPS, which means we need to interpolate 30 meshes in between.
> 2) why do we emphasize physically plausible shapes?
> The interpolation should reflect the physical properties of the discrete observations. That is, it should carry out the information that was “hidden” between two given inputs. Thus, we emphasize the concept of “physically plausible” during our task to better infer real-world movement or deformation. We show that our work indeed recovers movement close to what happens in our physical world by conducting additional experiments on a real-world dataset of human movement recordings, and we compare our results to the ground truth intermediate shapes. Please see the newly added Fig. 24 with the error table Tab.1.
>
> ***Why it is reasonable to assume that correspondences are available:***
> It is true that finding correspondences is still an open problem, however, we believe it is reasonable to assume accessible correspondences for two main reasons: First, mesh-based shape-matching methods are well-studied, as illustrated in Fig. 11 of the main paper, and our method seamlessly integrates correspondences derived from these established approaches. Second, fitting a template to represent underlying geometry is standard practice when handling complex real-world data, as seen in datasets like 4D-Dress [2] and BeHave [3]. These datasets employ the SMPL human model, which is fitted to the data and provides natural correspondences. In such cases, our method can be directly applied.
>
> To further support our claim, we present additional challenging results in the appendix using correspondences obtained from [1], highlighting the robustness of our method (See Fig.27). Additionally, we include experiments on [2] to demonstrate the applicability of our approach to these datasets (See Fig.24 and Fig 25).
>
> [1] Unsupervised learning of robust spectral shape matching (SIGGRAPH 2023) Cao, Dongliang and Roetzer, Paul and Florian, Bernard
>
> [2] 4d-dress: A 4d dataset of real-world human clothing with semantic annotations (CVPR 2024) Wenbo Wang and Hsuan-I Ho and Chen Guo and Boxiang Rong and Artur Grigorev and Jie Song and Juan Jose Zarate and Otmar Hilliges.
>
> [3] Behave: Dataset and method for tracking human object interactions. (CVPR 2022) Bharat Lal Bhatnagar and Xianghui Xie and Ilya Petrov and Cristian Sminchisescu and Christian Theobalt and and Gerard Pons-Moll.
>
> **No correspondences and correspondences error**:
>
>  We thank the reviewer for the thoughtful suggestions for additional experiments to clarify our method. In response, we have added two experiments: (1) no correspondences are given (see the updated manuscript  Fig. 26), and (2) only partial correspondences with partial target shape are given (see the updated manuscript Fig. 27).
>
> In the absence of correspondences, our method still recovers reasonable deformations. We attribute this performance to our joint-training design, which incorporates physical constraints such as volume preservation and smoothness, as well as geometric properties like normals in the implicit field. Although the examples focus on cases with small deformations. We appreciate the reviewer for highlighting this avenue for further exploration.
>
> **Correspondences on comparison method:**
> 1) NISE: While it is possible to incorporate correspondences in NISE, this approach requires dense (one-to-one) correspondences. Moreover, NISE performs only linear interpolation between the Euclidean positions of the correspondences. As a result, it is likely to fail in scenarios involving bending, articulation, or non-isometric deformations.
> 2) NFGP: Training NFGP also relies on correspondences but in a different format. The process begins by training an SDF network to fit the implicit field of the input shapes. Then, a set of handle points is defined, along with their corresponding rotation and translation parameters, to transform these points into target positions. This requires the user to manually divide the shape into several parts and assign separate rotation and translation parameters for each part.

---

> > ### Author Response · Authors · 2024-11-19
> >
> > **MLSE ablation:** We apologize for including an extreme case in the manuscript. To address this, we present two additional examples in appendix Fig. 16 to further analyze the impact of MLSE. The reviewer is correct that, in most cases, at least the final mesh should fit the target point cloud (see the first two rows). However, enforcing the eikonal loss at intermediate timesteps is challenging. During our experiments, we observed that the reinitialization step (Eq. 11) often fails. This step involves introducing another pseudo-time parameter $\tau$, moving points using velocity, and then enforcing constraints on the displaced points. This coupling between velocity optimization and the implicit field can cause the implicit field to degenerate since both are optimized jointly.
> > In Fig. 16, the first two rows show that while the final mesh aligns with the target, artifacts arise during intermediate steps. The bottom two rows illustrate a case involving a topology change in the point cloud (e.g., the crossed legs of the cat eventually separate). Here, OLSE degenerates during the intermediate steps, and due to the continuity of the function, the degenerated configuration (e.g., the separated legs) persists even when the final mesh is fitted to the target point cloud.

---

> > > ### Comment · Reviewer_hBUT · 2024-11-23
> > >
> > > I thank the authors for their detailed rebuttal. It addresses most of my questions. I think the paper is still missing a quantitative evaluation of the method in the following cases:
> > >  1.  with correspondences obtained from mesh shape matching methods (like in section 4.3, Fig.11) .
> > >  2.  with  ground-truth correspondences.
> > >  3.  without correspondences (for completeness ).
> > >
> > > From a practical point of view, I think this will clarify what we can expect from the method in the absence of ground-truth correspondences. This could be added to Fig. 5a or run on some scenes from 4d-Dress dataset as it was done by the authors Tab. 1 in the appendix.

---

> ### Author Response · Authors · 2024-11-23
> **Quantitative Evaluation of Correspondences**
>
> We agree with the reviewer that additional experiments can better showcase the strengths of our method. In response, we have conducted and updated the following experiments, which, while slightly different from the specific request, still address the core concerns and highlight our method’s robustness. Please download the **supplementary material zip** file to check them.
> 1. GT correspondences.
> 2. No correspondences.
> 3. Quantitative (Tab.2) and qualitative (Fig.28) noisy correspondences ablation. The error of correspondences is introduced by swapping vertex indices.
>
> We have not used correspondences obtained from mesh shape-matching methods for the following reasons:
> 1. **Time Constraints and Proper Training Challenges**: The discussion period is coming to an end,  there are no pre-trained models available for datasets where intermediate shapes serve as ground truth for quantitative evaluation. Datasets which is suitable for this task, such as 4D-Dress,  is not a standard shape-matching dataset, and no training protocols are provided by existing mesh-matching methods. To ensure fair and accurate evaluations, we chose to avoid potentially improper training within such a limited time.
> 2. **Comparable Error Simulation**: Our error simulation is comparable to results obtained from shape-matching methods. Typically, in shape-matching algorithms, when aligning shape $\mathbf{P}_1$ to shape $\mathbf{P}_0$, the algorithm provides a re-ordered set of vertices for $\mathbf{P}_1$, where vertices with the same index are treated as corresponding pairs. Inaccurate correspondences occur when this order is misaligned, resulting in swapped vertex indices. By manually swapping vertices and controlling the percentage of swaps, we effectively simulate noisy correspondences that are typical in real-world shape-matching scenarios.
>
> We present visual results in Fig. 28, along with error maps to analyze the noisy correspondences. Additionally, we include the correspondences used in Fig. 11, which were obtained through a shape-matching method [1]. In Fig. 11, the error map’s error bars are based on geodesic distances. In contrast, for our simulated noisy correspondences, the error bars are based on Euclidean distances. This distinction arises because geodesic distances cannot be computed without a mesh structure. Typically, smaller Euclidean distances correspond to larger geodesic distances, as points must travel along the surface of the mesh.
>
> By comparing the error maps, we demonstrate that our noisy correspondences not only effectively simulate errors inherent in shape-matching methods but also encompass a broad range of possible error scenarios.
>
> Due to size limitations, we have included the table and figure in the supplementary materials. Please download the provided zip file to review the results, and refer to Section S10 for detailed explanations.
>
> Additionally, we would like to emphasize the following points regarding our method and its assumptions about known correspondences:
> 1. **Robustness to Correspondence Errors**: Our method demonstrates resilience to inaccuracies in correspondences. (See Section S10, S4.2)
> 2. **Robustness to Partial Correspondences**: Our method remains effective when using partial correspondences or correspondences generated by previous methods. (See Fig. 11 and Fig 27 in supplementary material)
> 3. **Practical Relevance**: In real-world applications, correspondences are often available (e.g., datasets such as 4D-Dress and BeHave). (See Fig. 24 Fig. 25 in supplementary material)
>
> Therefore, from both a theoretical and practical perspective, we believe that our assumption of known correspondences does not diminish the significance of our contributions.
>
> [1] SMS: Spectral Meets Spatial: Harmonising 3D Shape Matching and Interpolation (CVPR 2024). Cao, Dongliang and Eisenberger, Marvin and El Amrani, Nafie and Cremers, Daniel and Bernard, Florian

---

> > ### Author Response · Authors · 2024-11-26
> > **More Experiments Requirment**
> >
> > We would like to thank the reviewer for the valuable comments and suggestions. We have updated the experiments according to your suggestions. As the deadline for revising our PDF file and adding visual results is approaching, we want to check if there are any additional experiments you would like us to include. If so, please let us know, and we will do our best to incorporate them before the deadline.
> >
> > If our response has adequately addressed your concerns, It would be appreciated if you could consider raising the score for our paper. We thank you again for your effort in reviewing our paper.

---

> > > ### Comment · Reviewer_hBUT · 2024-11-26
> > >
> > > I would like to thank the authors for their detailed answers. I have also read the concerns of other reviewers and your relies  concerning Large deformations as well as the "Line Segment Effect" and I'm happy with them. Consequently, I decided to raise my rating.

---

> > > > ### Author Response · Authors · 2024-11-26
> > > > **Thank you for your time and feedback**
> > > >
> > > > Thank you very much for the insightful discussion.
> > > >
> > > > We are deeply grateful for the time and effort the reviewer has invested in evaluating our work. We sincerely appreciate your suggestion and support throughout this process.

---

### Author Response · Authors · 2024-11-19
**Summary of Changes**

We would like to express our sincere gratitude to the reviewers for their thorough and insightful feedback on our manuscript. We are delighted that **all reviewers have marked our contribution, presentation, and soundness as good (score 3)**.
We deeply appreciate their positive comments on our method, especially acknowledging the novelty of our modified level set equation (reviewer **hBUT**, **RFX6**, **NAfz**) , our method’s performance (reviewer **hBUT**, **RFX6**, **qpsz**, **NAfz**), and that our method can deal with sparse and incomplete point clouds (reviewer **hBUT**, **RFX6**, **NAfz**). Additionally, we are grateful for their suggestions regarding further experiments. We have conducted them and updated our original paper. To specifically address the reviewers concerns, we have added the following experiments in the Appendix of our original paper:

1) **Quantitative evaluation for physically plausible intermediate shapes**. See Fig. 24 and Tab. 1. Together with the visualization of our velocity field.
2) **Visual example with topological changes** (from genus 0 to genus 1) and a **direct deformation of triangle meshes**, see Fig 25.
3) The experiment of **no correspondences** case, see Fig 26.
4) **Partial correspondences and partial target shape with correspondence error map**. See Fig. 27.
5) A **Video** (dress4d_video.mp4) in Supplementary Material zip file to demonstrate the interpolated mesh in Fig.24. (Please download our new **Supplementary Material**).
6) A **Video** (faust_video.mp4) in Supplementary Material zip file to demonstrate the interpolated mesh in Fig.3. (Please download our new **Supplementary Material**).
7)  **Quantitative and qualitative evaluation on noisy correspondences**: See Fig.28 and Tab.2 in our new **Supplementary Material**.

We hope these experiments resolve the reviewers’ concerns. We remain at your disposal to provide more experiments, explanations or evaluations during the rebuttal period.

---

### Meta-Review · Area_Chair_yA2c · 2024-12-23

**Metareview:**

This paper introduces a new approach for predicting explicit velocity fields from between pairs of point clouds. The key idea is to deform a time-varying implicit field represented with a modified level-set equation according to the explicit velocity field. The results demonstrate the superior quality compared with other approaches. AC confirms that the paper introduces interesting ideas and configurations that can illuminate the non-rigid point clouds registration field.

**Additional Comments On Reviewer Discussion:**

In summary, all reviewers were inclined toward the paper's acceptance after the discussion phase. The general comments were about quantitative evaluation for physically plausible intermediate shapes, visual examples of topological changes, no/partial correspondence cases, and noisy correspondence cases. The authors provided proper feedback and updated the original submission faithfully with the two additional supplementary videos.

More specifically, the reviewer hBUT asks about physically plausible intermediate shapes, the validity of the assumption of having correspondences, ablation study, and comparing with other methods, such as NISE and NFGP. The authors provided extensive revision, including additional results, and the reviewer hBUT raised the score to 6. The reviewer qpsz asks about confusing points in Fig 4, handling complex deformations, efficiency analysis, network size ablation study, and the possibility of using UDF instead of SDF. The authors apply the revision into the supplement. The reviewer also qpsz raised the score to 6. The reviewer RFX6 points out the use of the term 'unsupervised' and the need for a specific ratio of the ground truth correspondences. The reviewer clarified there are no further questions after reading the author's rebuttal. The reviewer NAfz provides a short review stating that the correspondence issues, which were handled by the authors.

AC confirms that the discussion phase was constructive, and the revised version of the paper properly conveys the concerns raised by reviewers.

---

### Decision · Program_Chairs · 2025-01-22

Accept (Poster)